# An independent poor-prognosis subtype of breast cancer defined by a distinct tumor immune microenvironment

Xavier Tekpli [1], Tonje Lien[1], Andreas Hagen Røssevold [2,3], Daniel Nebdal[1], Elin Borgen[4], Hege Oma Ohnstad [3], Jon Amund Kyte[2,3], Johan Vallon-Christersson [5], Marie Fongaard[1], Eldri Undlien Due[1], Lisa Gregusson Svartdal[4], My Anh Tu Sveli[4], Øystein Garred[4], OSBREAC, Arnoldo Frigessi[6], Kristine Kleivi Sahlberg[1,7], Therese Sørlie[1,8], Hege G. Russnes [1,4], Bjørn Naume[3,9] & Vessela N. Kristensen[1,8,10]*

How mixtures of immune cells associate with cancer cell phenotype and affect pathogenesis is still unclear. In 15 breast cancer gene expression datasets, we invariably identify three clusters of patients with gradual levels of immune infiltration. The intermediate immune infiltration cluster (Cluster B) is associated with a worse prognosis independently of known clinicopathological features. Furthermore, immune clusters are associated with response to neoadjuvant chemotherapy. In silico dissection of the immune contexture of the clusters identified Cluster A as immune cold, Cluster C as immune hot while Cluster B has a pro-tumorigenic immune infiltration. Through phenotypical analysis, we find epithelial mesenchymal transition and proliferation associated with the immune clusters and mutually exclusive in breast cancers. Here, we describe immune clusters which improve the prognostic accuracy of immune contexture in breast cancer. Our discovery of a novel independent prognostic factor in breast cancer highlights a correlation between tumor phenotype and immune contexture.

[1] Department of Cancer Genetics, Institute for Cancer Research, Oslo University Hospital, Oslo, Norway. [2] Department of Cancer Immunology, Institute for Cancer Research, Oslo University Hospital, Oslo, Norway. [3] Department of Oncology, Division of Cancer Medicine, Oslo University Hospital, Oslo, Norway. [4] Department of Pathology, Division of Laboratory Medicine, Oslo University Hospital, Oslo, Norway. [5] Division of Oncology and Pathology, Department of Clinical Sciences Lund, Faculty of Medicine, Lund University, Scheelegatan 2, Medicon Village, 22185 Lund, Sweden. [6] Department of Biostatistics, Oslo Centre for Biostatistics and Epidemiology, University of Oslo and Research Support Services, Oslo University Hospital, Oslo, Norway. [7] Department of Research, Vestre Viken Hospital Trust, Drammen, Norway. [8] Centre for Cancer Biomarkers CCBIO, Bergen, Norway. [9] Institute of Clinical Medicine, University of Oslo, Oslo, Norway. [10] Department of Clinical Molecular Biology, Division of Medicine, Akershus University Hospital, Lørenskog, Norway. A full list of consortium members appears at the end of the paper. *email: Vessela.N.Kristensen@rr-research.no

The tumor microenvironment influences cancer initiation and progression[1,2]. In breast cancer, clinicopathological characteristics such as age, grade, stage, and molecular subtypes associate with prognosis and drive treatment decisions. High-throughput gene expression analyses led to a molecular classification of breast cancers[3,4]. The five clinically relevant molecular subtypes: Luminal A, Luminal B, Her2-enriched, Basal-like, and Normal-like, have different incidences, survival, prognosis, and tumor biology. Such patient stratification has clinical and economical utility in breast cancer management[5].

In addition to cancer cell biology, an inflammatory micro-environment influences initiation and progression[6]. The immune microenvironment surrounding cancer cells can recognize and inhibit tumor growth[7] or promote progression[8]. It is crucial to characterize the quality and quantity of immune response at the tumor site, as it may help to pinpoint patients who could benefit from immunotherapies and will improve our understanding of the tumor–host biology.

In breast cancer, high immune infiltration has been associated with better clinical outcome[9,10]. In particular, high CD8+ T cell infiltration associate with better overall survival (OS) in estrogen receptor (ER)-negative patients[11,12]. In addition, high immune infiltration has been associated with an increased response to neo-adjuvant and adjuvant chemotherapy[13].

Recently, we and others have demonstrated that transcriptomic data can be leveraged to dissect the tumor microenvironment[14–19]. Such methods have shown that elevated expression of leukocyte marker genes associates with a lower risk of breast cancer recurrence[14,17,20,21]. Notably, Ali et al. and Bense et al. recently reported through comprehensive studies how specific immune cell types influence breast cancer outcome[14,22]. In these studies, the authors assessed each predicted cell type individually and did not consider the immune microenvironment as a whole. More studies are needed to specify the role and the clinical relevance of the immune contexture in breast cancer.

In the present study, we discover clinically relevant immune clusters with gradual immune infiltration. In 15 breast cancer cohorts, spanning 6101 breast cancer samples, the group of patients with intermediate levels of tumor immune infiltration has a worse prognosis independently of known prognostic molecular and clinicopathological features. Through characterization of the immune composition of the clusters, we find a pro-tumorigenic immune infiltration associated with the poor prognosis group. Further phenotypical analyses show two mutually exclusive aggressive tumor phenotypes in breast cancers, one linked to epithelial–mesenchymal transition (EMT) and the other to proliferation. Both phenotypes are found in the poor prognosis cluster on an inactive/pro-tumorigenic immune microenvironment.

## Results

**Immune clusters in breast cancer**. The expression of 760 genes in 95 formalin-fixed, paraffin-embedded (FFPE) tumor samples of the MicMa cohort was measured using the nCounter® Pan-Cancer Immune Profiling array, an array designed to profile immune infiltration in solid tumors. Seventy-nine of these 95 samples have been previously profiled by Agilent whole-genome 4 × 44K oligo array[23]. We first compared the expression obtained with the two platforms using Pearson and Spearman correlations and found a high degree of positive correlation between the genes' expression values (Supplementary Fig. 1A).

In order to group patients according to their similarity in expression of the immune-related genes, we performed unsupervised hierarchical clustering of the correlation matrix (Fig. 1a: 95 MicMa-nCounter and Supplementary Fig. 1B; 104 MicMa-Agilent samples). Silhouette plot analysis from 3 to 10 clusters

indicated that 3 clusters captured best the segmentation of both the nCounter and the Agilent datasets (Supplementary Fig. 1C, D). We therefore continued our analyses based on three clusters of patients. We compared the clustering obtained from FFPE: MicMa-nCounter, 95 samples (correlation matrix obtained from the expression of 760 genes on the Immune Profiling array) to the clustering performed on fresh frozen tissue MicMa-Agilent, 104 samples (correlation matrix obtained from the expression of the 509 genes on the Immune Profiling array found in all datasets used in this study). Seventy-nine samples were overlapping in these two datasets. With different platforms used to measure gene expression, as well as incomplete overlap in gene lists and samples used to perform unsupervised clustering, we still found the cluster assignment for the 79 overlapping samples significantly similar (Supplementary Table 1 with Fisher exact test <0.0001).

To confirm that the clusters were associated with the tumor immune microenvironment (Fig. 1b), we used the algorithm Nanodissect to score for total lymphocyte and myelocyte infiltration[17,24,25]. Nanodissect scores were first validated in the MicMa cohort using the evaluation of immune infiltration of matched hematoxylin and eosin (H&E) sections analyzed by experienced pathologists (Fig. 1c and Supplementary Fig. 1E).

We found the three clusters significantly correlated with Nanodissect lymphocyte (Fig. 1b) and myelocyte (Supplementary Fig. 1F) scores. In addition, Chi-squared test showed significant association between clusters and immune infiltration assessed by experienced pathologists ($p < 0.0001$). We concluded that Clusters A–C reflect gradual immune infiltration and were therefore called immune clusters.

**Clusters reflect gradual immune infiltration**. We validated the association between the clusters and lymphoid/myeloid infiltration using the expression data from nine other cohorts (Supplementary Table 2). As stated above, 509 of the 760 genes on the nCounter® PanCancer Immune Profiling array were found in all datasets studied, the expression of these 509 genes was used in the unsupervised clustering (Fig. 1d and Supplementary Fig. 2A for the clustering of the METABRIC and The Cancer Genome Atlas (TCGA) cohorts, respectively). In each cohort, the three clusters obtained were significantly associated with lymphoid and myeloid Nanodissect scores (Lymphoid score: METABRIC, Fig. 1e; TCGA, Supplementary Fig. 2B).

Lymphoid and myeloid infiltrations gradually increased from Cluster A (blue; low infiltration; cold tumors) to Cluster B (light blue; intermediate infiltration) and Cluster C (pink; high infiltration; hot tumors).

For an additional layer of validation, we used the pathological assessment of immune infiltration in the METABRIC cohort[26], which was significantly associated with the Nanodissect scores (Fig. 1f and Supplementary Fig. 2C) and with the immune clusters: Chi-square test between immune clusters and pathological assessment of immune infiltration $p$ value < 0.0001. We could now strongly conclude that unsupervised hierarchical clustering using genes of the PanCancer Immune Profiling array allows to group breast cancer tumors according to gradual levels of immune infiltration.

**Immune clusters associate with prognosis**. We examined the immune clusters in perspective of survival using Kaplan–Meier analysis and log-rank tests. For the two largest cohorts METABRIC ($n = 1904$) and TCGA ($n = 981$), we found Cluster B (with intermediate levels of immune infiltration) associated with worse prognosis (Supplementary Fig. 3A, B). Such a worse outcome for Cluster B cases was also observed when stratifying for ER-negative (Supplementary Fig. 3C, D) and ER-positive cases

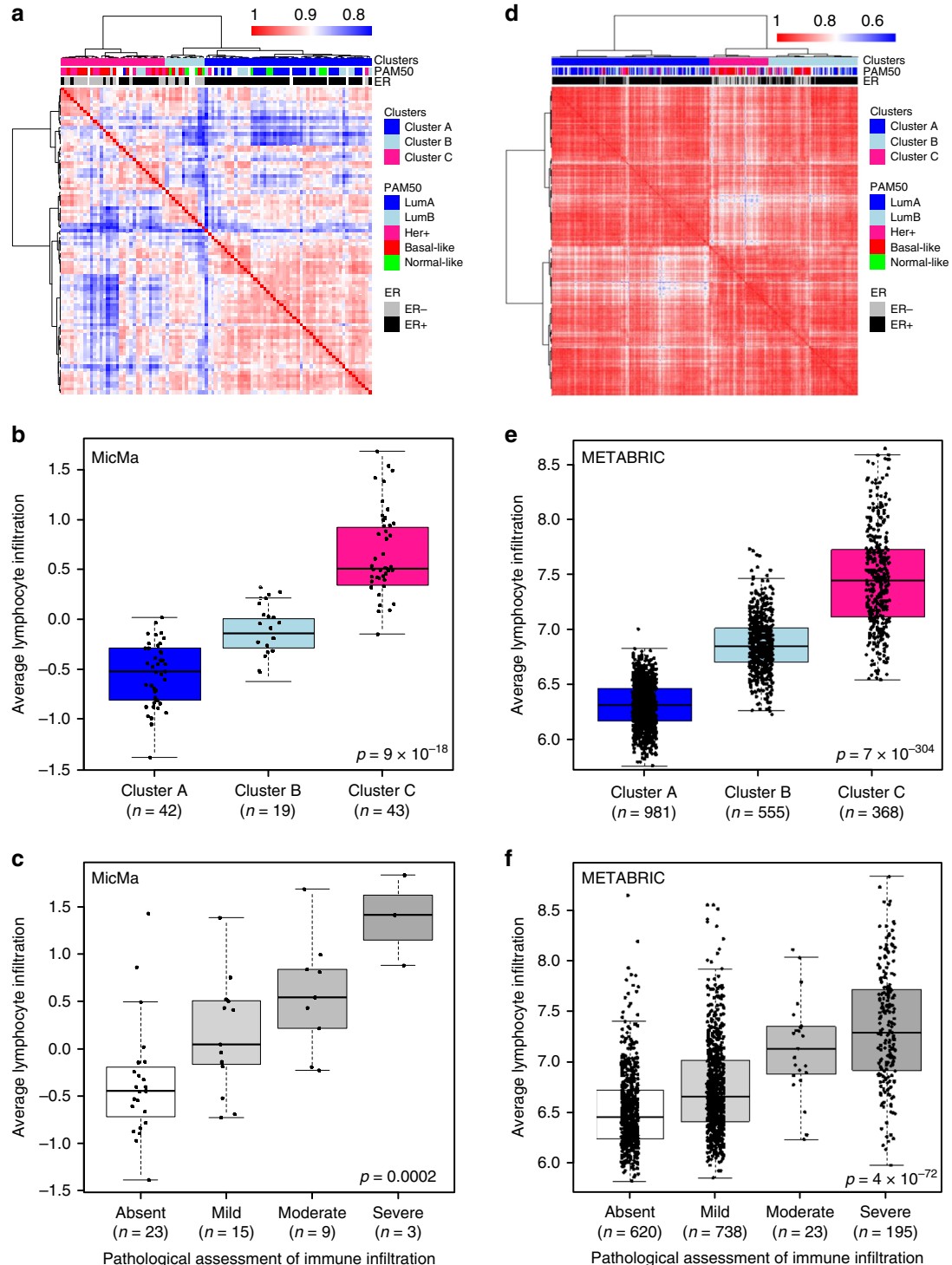

**Fig. 1** Immune clusters are associated with total immune infiltration. **a**, **d** Gene expression was measured in 95 FFPE MicMa (**a**) and 1904 fresh frozen METABRIC samples (**d**). Unsupervised clustering using correlation distance and ward. D linkage of the correlation matrix assesses the relation between patients according to the expression of the genes on the PanCancer Immune Profiling array. All 760 genes on the array were used for clustering the MicMa cohort, while 509 genes, which corresponds to genes (out of the 760) found in all datasets, were used to cluster the METABRIC. Annotations of the samples on the top of the heatmap indicate histopathological features: PAM50 subtype, ER status as well as the three clusters identified by the cutree method. **b**, **e** In the MicMa (**b**) and the METABRIC (**e**), lymphoid scores quantify lymphoid infiltration which was calculated from a set of genes' markers of lymphocyte as defined by the algorithm Nanodissect. Lymphoid scores are represented in boxplots according to immune clusters with Kruskal–Wallis test $p$ values. **c**, **f** H&E-stained tumor tissue samples (**c**, MicMa, $n = 50$ and **f**, METABRIC, $n = 1904$) were categorized by an experienced pathologist according to the level of tumor-infiltrating immune cells. Boxplots represent the average lymphocyte score from Nanodissect according to pathologists' classifications. Kruskal–Wallis test $p$ values is denoted. The line within each box represents the median. Upper and lower edges of each box represent 75th and 25th percentile, respectively. The whiskers represent the lowest datum still within [1.5 × (75th − 25th percentile)] of the lower quartile, and the highest datum still within [1.5 × (75th − 25th percentile)] of the upper quartile.

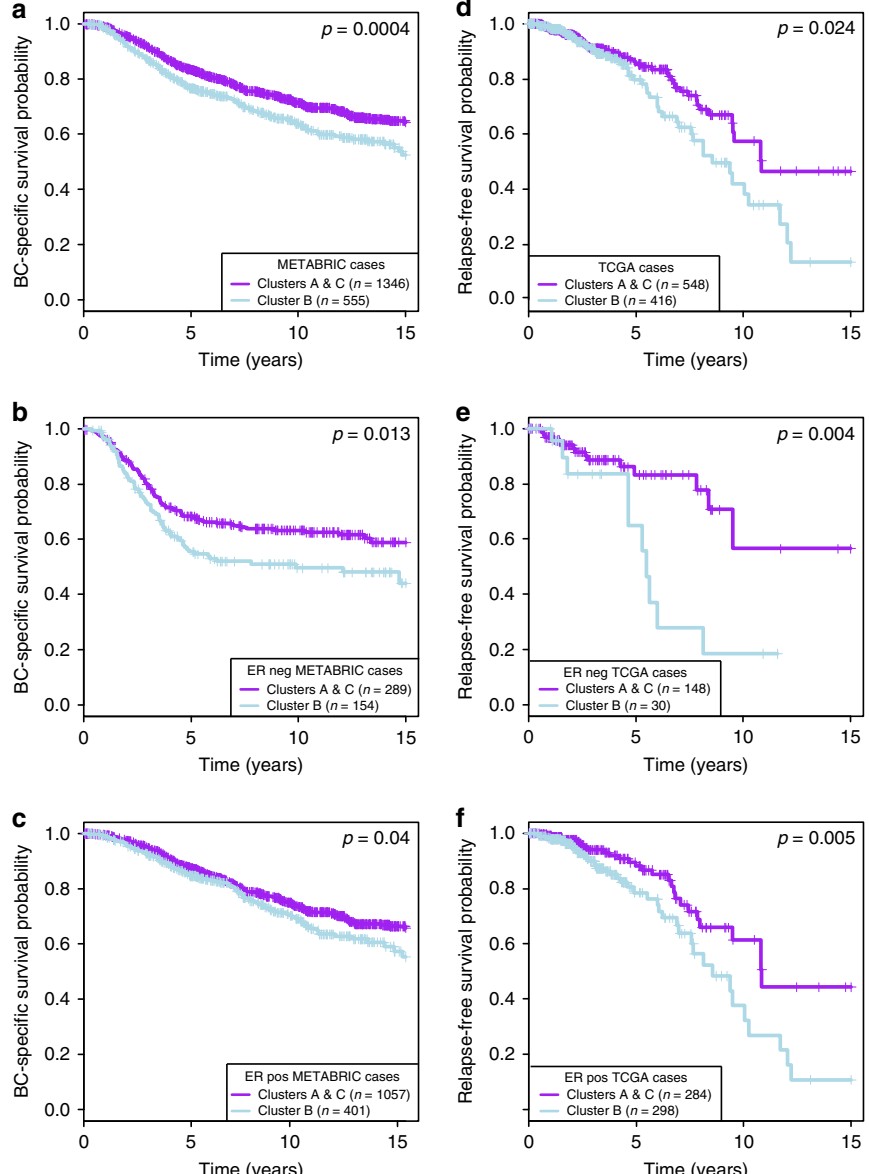

**Fig. 2** Immune clusters are associated with prognosis. Kaplan–Meier survival curves for Cluster B (light blue) and Clusters A and C (purple). In all METABRIC (**a**) and TCGA (**d**) samples; in ER negative (**b**, **e**) and ER positive (**c**, **f**). p values are from log-rank tests. Kaplan–Meier display breast cancer-specific survival for the METABRIC and relapse-free survival for the TCGA.

(Supplementary Fig. 3E, F) separately. To refine our observation, we plotted patient survival according to Cluster B (light blue) vs Clusters A and C (purple) and confirmed a clear and significant worse prognosis for patients in Cluster B (Fig. 2). We further validated this result in four additional cohorts with relevant survival data: TAI ($n = 327$), VDX ($n = 344$), STK ($n = 159$), and UPP ($n = 251$) (Supplementary Fig. 4). We concluded that immune clusters associate with prognosis both in ER-negative and ER-positive breast cancers.

**Predicting immune clusters with binomial logistic regression**. Motivated by the clinical relevance of the immune clusters, we aimed at developing a general method that could precisely and sensitively predict the classification of patients to the worse prognosis group without having to rely on unsupervised clustering. We developed a model through training on 10 cohorts (4546 samples) and testing on 5 others (1555 samples). We used

binomial logistic regression penalized by the lasso method to obtain a set of genes (Supplementary Data 1) that sensitively and specifically predict whether a sample is part of Cluster B or not, as assessed by receiver operating characteristic curve and area under the curve (AUC) analysis (Fig. 3a). Our model predicted the immune clusters with an AUC = 85.8% (82.8%–88.7%). We found that 96.3% of the samples assigned to Clusters A and C by clustering were predicted to be A and C by the model, while 68.8% of the samples assigned to Cluster B through clustering were found in Cluster B using the lasso method (Fig. 3b). It appeared that the lasso method decreased the number of samples in Cluster B (Fig. 3b). As unsupervised clustering is less reliable in small cohorts and because learning the cluster assignment from several cohorts will help to precise the phenotype underlying the immune clusters, we hypothesized that the lasso-derived classification would be a better prognostic factor than the clustering method. Indeed, by comparing the survival log-rank test p values, we found that the lasso classification generally improved the

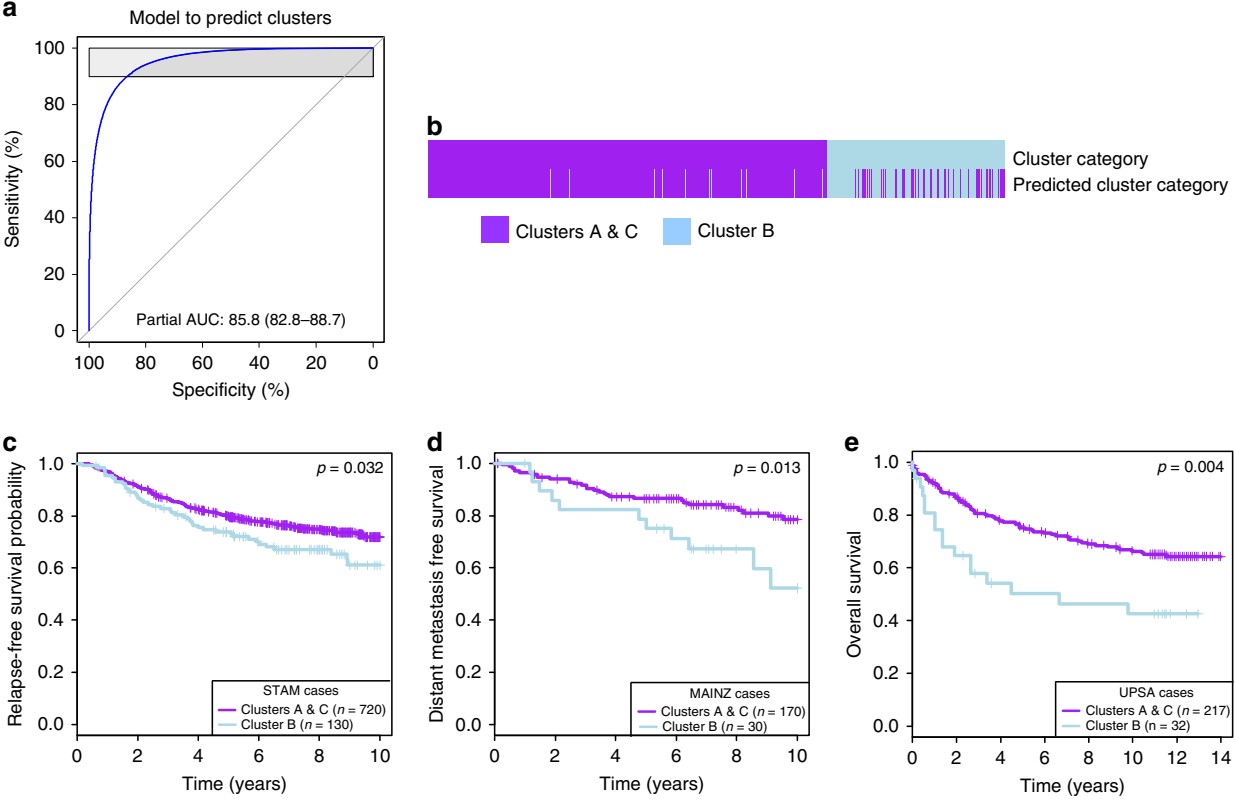

**Fig. 3** Prediction of Cluster B using binomial logistic regression. **a** Using binomial logistic regression penalized by the lasso method, we trained on 4546 samples to predict Cluster B. ROC curve assesses how the lasso output (the weighted gene sets in Supplementary Data 1) discriminates a sample to be Cluster B or not. **b** For the 4546 samples in the training set, the heatmap represents whether a sample is part of Cluster B (light blue) or Clusters A and C (purple), using the clustering or the lasso methods. **c–e** The prediction of the clusters (lasso) was tested on five cohorts, which were not included in the training phase: **c** STAM ($n = 856$), **d** MAINZ ($n = 200$), and **e** UPSA ($n = 289$) are presented here, the two other cohorts CAL and PNC are presented in Supplementary Figures. The association between predicted clusters and survival was tested using Kaplan–Meier survival curves for predicted Cluster B (light blue) and predicted Clusters A and C (purple). $p$ values are from log-rank tests. Kaplan–Meier display relapse-free survival for STAM, distant metastasis-free survival for MAINZ, and overall survival for UPSA.

significant associations between the immune clusters and survival (Supplementary Table 3). The lasso model was validated in five additional cohorts: Fig. 3c–e for STAM ($n = 856$), MAINZ ($n = 200$), and UPSA ($n = 289$) and Supplementary Fig. 5A, B for CAL ($n = 118$) and PNC ($n = 92$).

As the binomial logistic regression only predicted two clusters (Cluster B vs Clusters A and C), we performed another round of binomial logistic regression to distinguish between Cluster A and C with high accuracy (Supplementary Fig. 5C, D). In conclusion, binomial logistic regression penalized by the lasso method refined Cluster B and provided a single sample predictor that could be applicable to every next patient in the clinic. In the subsequent analyses, we use the categories given by the lasso methods as it has a more significant association with survival.

**Immune clusters, an independent prognostic factor.** We further investigated how the immune clusters were related to well-known clinicopathological features in breast cancer (size, age, grade, stage, lymph node involvement, and molecular subtypes (PAM50)). Cluster A (with low immune infiltration) was enriched in ER-positive and Luminal cases, while a higher proportion of ER-negative and Basal-like cases was found in Cluster C (with high immune infiltration) (Fig. 4a, b). ER-negative and ER-positive samples as well as the PAM50 subtypes were equally represented in the poor prognosis Cluster B (Fig. 4a, b).

We tested the prognostic impact of the immune clusters while accounting for other prognostic factors using multivariable Cox

regression analysis. The variables available for each cohort (ER status, PAM50 subtypes, age, nodal status, size, and grade) were entered into each model. The odd ratios and $p$ values associated with each variable in each model are shown in Supplementary Table 4. We found that immune clusters were an important factor to model survival as shown by the significant $p$ values associated with immune clusters in each cohort Cox model. Indeed, if we removed the immune clusters from the modeling, the Akaike Information Criterion (AIC) index was increased (Supplementary Table 5), demonstrating the important value of immune clusters on top of all other variables for explaining breast cancer survival.

To further test the strength of the immune clusters as an important prognostic biomarker, we used a stepwise backward selection. From the initial Cox models containing all variables, we removed the weakest predictor variable only if this did not weaken the model (as monitored by the calculation of AIC index). This allowed us to find for each cohort the set of variables explaining survival best. For all cohorts, the immune clusters were kept in the best fitted minimal model, and in 9 out of 11 cohorts, the immune clusters were a significant prognostic variable (Table 1). To further emphasize and illustrate the clinical relevance of the immune clusters and their independence from the PAM50 molecular subtypes, we plotted for the METABRIC and TCGA cohorts the Kaplan–Meier survival curve for each PAM50 subtype (Supplementary Fig. 6).

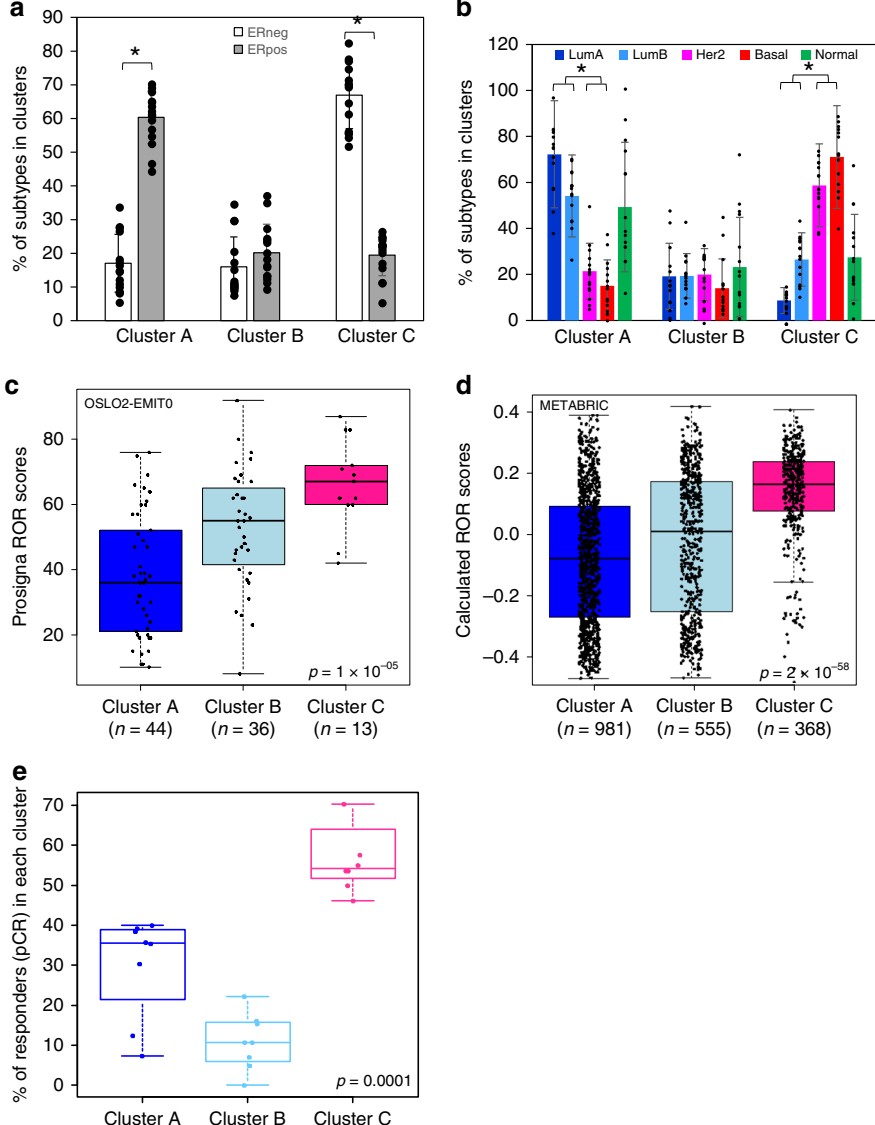

**Fig. 4 Immune clusters and clinicopathological features. a, b** Average percentage of ER-positive and ER-negative samples (**a**) or PAM50 subtypes (**b**) of 15 cohorts across the clusters. Cluster A is enriched for ER-positive, Luminal A, Luminal B samples while a significantly higher percentage of ER-negative, Basal-like, Her2-enriched samples was found in Cluster C (high infiltration). Asterisk (*) denote *t* test *p* value < 0.0001. Error bars represent standard error to the mean. **c** Prosigna Breast Cancer ROR scores for the OSLO2-EMIT0 cohort were obtained from the NanoString nCounter Dx Analysis System using FFPE breast tumor tissue. Boxplots represent the average Prosigna ROR scores in the immune clusters. Kruskal–Wallis test *p* value is denoted. **d** Calculated ROR scores following the method of Parker et al.[3] are compared to immune clusters using boxplots in the METABRIC cohort. Kruskal–Wallis test *p* values is shown. **e** From eight breast cancer cohorts, in which the pathological complete response (pCR) was assessed after administration of neoadjuvant chemotherapy, we calculated the percentage of responders in each cluster. Boxplots show the distribution of the percentage of responders in each immune cluster. Kruskal–Wallis test *p* value is denoted. The line within each box represents the median. Upper and lower edges of each box represent 75th and 25th percentiles, respectively. The whiskers represent the lowest datum still within [1.5 × (75th − 25th percentile)] of the lower quartile and the highest datum still within [1.5 × (75th − 25th percentile)] of the upper quartile.

**Validation in a new RNA-seq dataset with risk of recurrence (ROR) scores**. We generated a new dataset: EMIT0, which is a subset of the OSLO2 cohort study. The OSLO2-EMIT0 was assessed by the Food and Drug Administration-approved Prosigna risk of recurence (ROR) scores. As recently demonstrated, ROR scores add significant prognostic information above standard clinicopathological features[3,27]. We assessed whether the immune clusters could add prognostic value to ROR scores. We found Cluster B composed of samples with intermediate ROR scores compared to Clusters A and C (Fig. 4c). This suggested that the poor prognosis associated with Cluster B was not likely to be explained by the information contained in the ROR scores.

This observation was also true when assessing the ER-negative (Supplementary Fig. 7A) and ER-positive (Supplementary Fig. 7B) cases separately. For all cohorts, we calculated the ROR scores following Parker et al.[3]'s method, which is related to PAM50 subtyping[3], and confirmed that Cluster B was composed of intermediate ROR scores as exemplified in the METABRIC cohort (Fig. 4d and Supplementary Fig. 7C, D).

Multivariable regression analysis confirmed that the immune clusters bring additional prognostic value to the ROR scores (Supplementary Table 6) as demonstrated by the significant *p* values for the immune clusters when modeling survival with ROR scores and immune clusters. Through computation of net

**Table 1 Summary statistics of the Cox regression analysis and stepwise backward selection.**

| | TCGA | METABRIC | TAI | VDX | STK | UPP | MAINZ | STAM | UPSA | CAL | PNC |
|---|---|---|---|---|---|---|---|---|---|---|---|
| **Immune clusters** | | | | | | | | | | | |
| Cluster B vs Clusters A–C | 0.38 (0.24–0.61) <0.0001 | 0.72 (0.59–0.88) 0.001 | 0.63 (0.37–1.05) 0.078 | 0.47 (0.32–0.7) <0.0001 | 0.47 (0.22–1) 0.049 | 0.51 (0.29–0.88) 0.02 | 0.34 (0.16–0.7) 0.004 | 0.65 (0.47–0.92) 0.014 | 0.57 (0.33–1.01) 0.053 | 0.22 (0.09–0.54) 0.001 | 0.55 (0.24–124) 0.151 |
| **ER status** | | | | | | | | | | | |
| ER neg vs ER pos | 0.54 (0.33–0.89) 0.015 | | | | | | | | 2.37 (1.04–5.36) 0.039 | | |
| **PAM50** | | | | | | | | | | | |
| Basal-like vs Her2 | | 1.14 (0.86–1.5) 0.370 | 2.29 (1.02–5.1) 0.044 | 0.9 (0.49–1.65) 0.726 | 1.67 (0.6–4.67) 0.326 | | 0.43 (0.14–1.35) 0.147 | 0.84 (0.46–1.53) 0.572 | 1.7 (0.72–4.03) 0.227 | | 2 (0.66–6.07) 0.222 |
| Basal-like vs LumA | | 0.41 (0.31–0.54) <0.0001 | 0.81 (0.34–1.93) 0.636 | 0.63 (0.37–1.08) 0.093 | 0.26 (0.09–0.78) 0.016 | | 0.19 (0.07–0.5) 0.001 | 0.39 (0.25–0.61) <0.0001 | 0.4 (0.16–1.04) 0.061 | | 0.62 (0.14–2.68) 0.526 |
| Basal-like vs LumB | | 0.92 (0.72–1.19) 0.529 | 1.58 (0.71–3.53) 0.262 | 1.39 (0.85–2.26) 0.188 | 1.19 (0.53–2.68) 0.673 | | 0.46 (0.19–1.1) 0.081 | 0.95 (0.64–1.41) 0.792 | 0.97 (0.38–2.5) 0.953 | | 1.18 (0.42–3.33) 0.750 |
| Basal-like vs Normal | | 0.72 (0.5–1.04) 0.079 | 2.69 (1–7.22) 0.050 | 0.25 (0.06–1.06) 0.060 | 0.16 (0.02–1.28) 0.084 | | 0.54 (0.18–1.63) 0.275 | 0.07 (0.01–0.48) 0.007 | 0.62 (0.2–1.89) 0.400 | | 7.58 (0.86–66.95) 0.068 |
| **Lymph node status** | | | | | | | | | | | |
| Node neg vs Node pos | | 2.15 (1.74–2.65) <0.0001 | | | | 2.59 (1.51–4.46) <0.0001 | | 1.64 (1.2–2.23) 0.002 | | | 1.35 (0.63–2.92) 0.443 |
| **Stage** | | | | | | | | | | | |
| Stage 1 vs Stage 2 | 1.83 (0.91–3.69) 0.088 | 0.73 (0.58–0.92) 0.008 | | | | | | | | | |
| Stage 1 vs Stage 3 | 2.2 (1.05–4.63) 0.037 | 1.38 (0.97–1.96) 0.073 | | | | | | | | | |
| Stage 1 vs Stage 4 | 5.68 (2.18–14.82) <0.0001 | 3.09 (1.43–6.68) 0.004 | | | | | | | | | |
| **Age** | | | | | | | | | | | |
| Continuous variable | 1.03 (1.02–1.05) <0.0001 | 1.01 (1–1.01) <0.0001 | | | | | | | | | |
| **Tumor size** | | | | | | | | | | | |
| Continuous variable | | 1.01 (1–1.01) <0.0001 | | | | 1.53 (1.24–1.89) <0.0001 | | | 1.01 (1–1.02) 0.040 | 0.86 (0.68–1.09) 0.209 | 1.09 (0.89–1.33) 0.406 |

Hazard ratio and 95% confidence interval are on the first row, p values from Cox proportional hazards model on the second row
For each cohort, the best fitted model (variables independently explaining survival) are shown

reclassification improvement (NRI) and integrated discrimination improvement (IDI) indexes[28], we emphasized the additional value of immune clusters to classify patients according to survival when taken together with ROR scores, as indicated by the positive NRI and IDI coefficients in all cohorts. Bootstrapping for confidence interval (CI) construction for NRI and IDI showed that, for several cohorts, the immune clusters significantly improved patient classification according to prognosis when added to the ROR scores (Supplementary Table 6). Using complementary statistical analyses, we demonstrate the clinical relevance of the immune clusters in breast cancer.

**Immune clusters and response to neoadjuvant chemotherapy.** We further assessed the association between the immune clusters and response to neoadjuvant chemotherapy, using gene expression data from studies in which patients were treated in neoadjuvant setting (chemotherapy before surgery). The endpoint of these studies was pathological complete response (pCR), which means complete eradication of cancer cells at the end of the chemotherapeutic regimen before surgery (see Supplementary Table 2 for datasets used in this section). We used gene expression data from 8 studies (1377 samples), and assigned to each sample its immune cluster belonging using the lasso method. As shown in Fig. 4e, we found the highest percentage of responders in Cluster C (59%), followed by Cluster A (30%) and the lowest percentage of responders in Cluster B (11%). Since Cluster B is also the smallest cluster in terms of patient numbers, we also calculated the percentage of responders within each cluster. Cluster C was composed in average of 42% of responders and 58% of patients with residual disease, whereas Cluster B had 18%/82% and Cluster A 13%/87% of responders/residual disease cases, respectively.

As the pCR rate differs as a function of ER status[29], we also calculated the percentage of responders in ER-positive and ER-negative cases independently and found the lowest rate of responders in Cluster B regardless of ER status (Supplementary Fig. 8A, B, respectively).

For each cohort with response to neoadjuvant chemotherapy, we assessed the distribution (chi-square p values, Supplementary Table 7) of the pCR and non-pCR cases across the immune clusters taking into account all cases, or ER-positive and ER-negative cases independently. When considering the whole cohort, we found the distribution of the responders significantly different across immune clusters, with less responders in Cluster B and most responders in Cluster C. When splitting by ER status, the same tendency was observed although not always significant.

These results demonstrate that patients in Cluster C have a higher probability to be responders, which corroborate previous studies reporting a higher pCR rate for cases with high immune infiltration and/or proliferative phenotype[29,30]. Our results also highlight a low response rate in Cluster B, suggesting that such patients may be candidates for testing of new neoadjuvant therapeutic options.

**In silico dissection of the immune clusters.** To assess whether the gradual immune infiltration in the clusters could explain the association with prognosis, we tested which of the immune clusters or total immune infiltration scores was more predictive of survival in a Cox multivariable regression analysis (Supplementary Table 8). Nanodissect lymphocyte scores were poorly associated with survival, we therefore hypothesized that specific immune cell-type mixtures, rather than the total number of immune cells in the tumor microenvironment, may explain the poor prognosis in Cluster B.

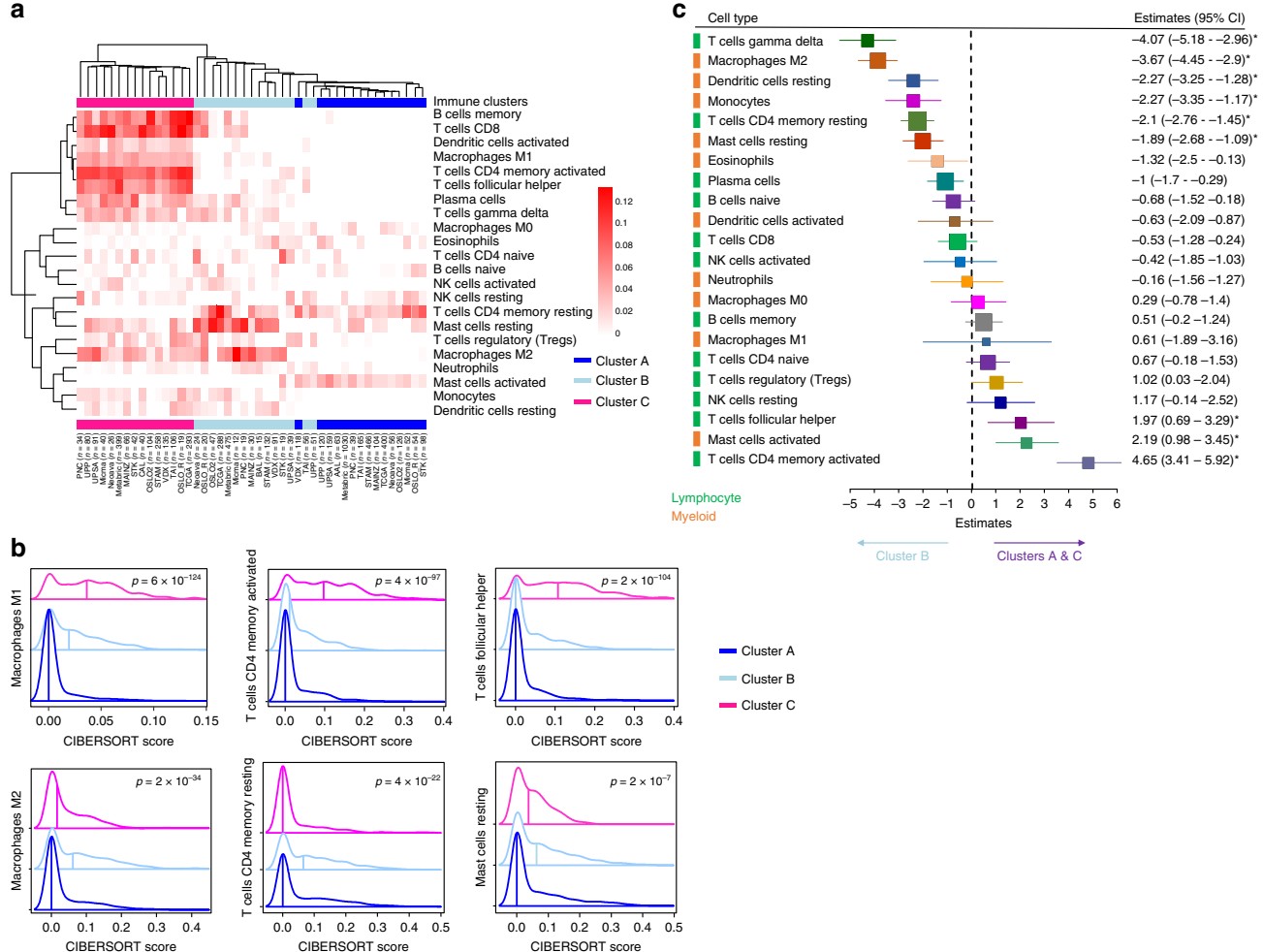

**Fig. 5** In silico dissection of the immune clusters. **a** We used the CIBERSORT algorithm to assess the composition of the immune microenvironment of breast cancer samples. For each cluster, we calculated the median of the absolute score of the 22 cell types given by the CIBERSORT in each cohort. Immune cluster-specific cell-type median scores were used in an unsupervised clustering using maximum linkage and the ward.D2 method. The heatmap obtained allows to visualize which cell types are enriched across the immune clusters. Immune clusters are annotated on the top and bottom of the heatmap. **b** Density plots represents the distribution of the absolute CIBERSORT scores for selected cell types across the clusters for the METABRIC cohort; the vertical lines crossing the distribution identify the median value for the score. Kruskal–Wallis test $p$ value are denoted. **c** Estimates of multivariable logistic regression analysis and the 95% confidence interval (CI) are illustrated by forest plot to assess which immune cells inferred by CIBERSORT explain the most the poor prognosis cluster (Cluster B) vs Clusters A and C. Box size is inversely proportional to the width of the confidence interval. Asterisks denote FDR-corrected $p$ value < 0.05. Immune cell types from the lymphoid or myeloid lineage are identified.

We estimated the proportions of 22 distinct immune cell types using the CIBERSORT algorithm[19]. We calculated *per* cohort and cluster the median infiltration of each immune cell type and performed unsupervised clustering of such cell-type-specific median infiltration scores (Fig. 5a). We found that the CIBERSORT inferred immune infiltration recapitulated the immune clusters. Cluster C cases were enriched, among other cell types, for macrophages M1, memory activated T cells, and follicular T helper cells (Fig. 5a), as also illustrated by the distribution of the CIBERSORT scores in the METABRIC and the TCGA cohorts (Fig. 5b and Supplementary Fig. 9). Cluster A had, as expected, very low levels of immune cells. In the poor response and prognosis Cluster B, higher levels of macrophages M2, resting mast cells, and resting memory T cells were found (Fig. 5a), as also illustrated by density plots for the METABRIC and TCGA cohorts (Fig. 5b and Supplementary Fig. 9).

Using generalized linear models, we specified the immune cell types distinguishing between Cluster B vs A–C and identified resting and pro-tumorigenic immune cell types explaining Cluster B

(Fig. 5c). We also tested which immune cell types explained the differences between Cluster A versus Cluster B (Supplementary Fig. 10A) and between Cluster B versus Cluster C (Supplementary Fig. 10B). When comparing Cluster A to Cluster B, all immune cell types could explain Cluster B, indeed, Cluster A has no or low immune infiltration. When comparing Cluster B to C, we found again the pro-tumorigenic cell types macrophages M2 and resting mast cells explaining Cluster B. These results suggest that pro-tumorigenic immune infiltration in Cluster B may favor tumor growth. In conclusion, Cluster A is composed of immune-cold tumors, Cluster C contains immune-hot tumors, and cases in Cluster B have a pro-tumorigenic immune infiltration.

**Phenotypic analysis of the immune clusters**. To further characterize the phenotype associated with the poor prognosis in Cluster B, we identified through differential gene expression analysis the genes significantly overexpressed in Cluster B. We found 909 genes upregulated in Cluster B when compared to

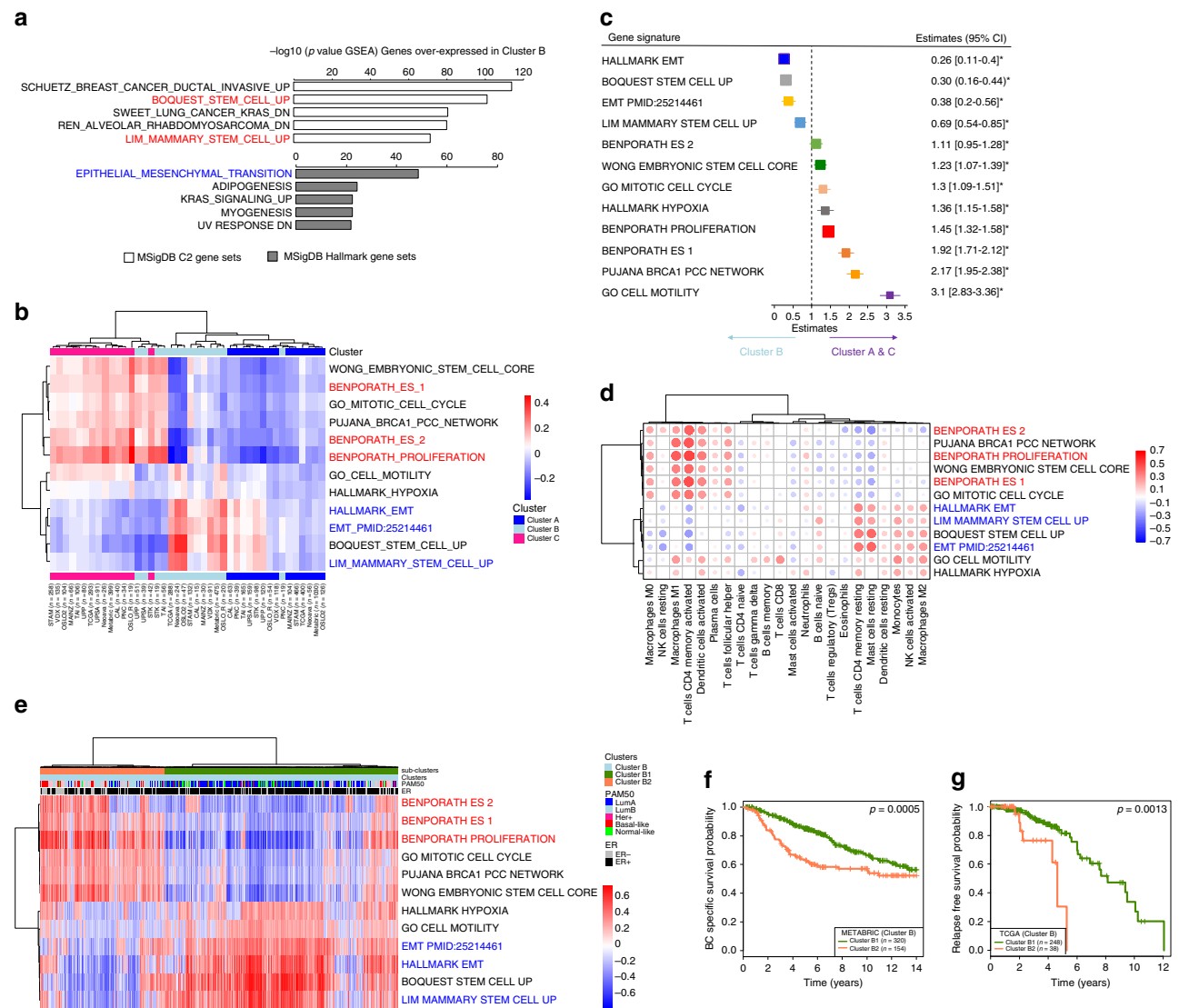

**Fig. 6** Immune clusters are associated with EMT and proliferation, two mutually exclusive phenotypes in breast cancer. **a** Genes overexpressed in Cluster B were defined using Bonferroni-corrected differential expression analysis (Cluster B vs Cluster A and Cluster B vs Cluster C). Genes with significantly higher expression in Cluster B were used in a gene set enrichment analysis using the C2 (white histograms) and H (gray histograms) collections of the MsigDB. −Log10 $p$ value of hypergeometric test are presented. The five most enriched processes in each collection are denoted. **b** Samples from each cohort (15 cohorts; 6101 samples) were scored using the GSVA Bioconductor package for enrichment in 12 pathways related to proliferation, EMT, and stem cells (Supplementary Data 3). Average enrichment scores are calculated per immune cluster and cohort. Unsupervised clustering using maximum method and ward. D2 linkage shows that pathways enrichment scores recapitulate the immune clusters. The numbers of samples in each cohort and immune clusters are denoted. Immune cluster from which the median score originate are annotated. **c** Estimates of univariate logistic regression analysis and the 95% confidence interval (CI) are illustrated by forest plot to assess which gene set signature scores calculated using GSVA associate with the poor prognosis cluster (Cluster B) vs Clusters A and C. Box size is inversely proportional to the width of the confidence interval. Asterisks denote FDR-corrected $p$ value < 0.05. **d** Correlation plots represent all the significant (FDR $p$ value < 0.05) Spearman correlations between gene set signature scores and inferred immune infiltration at the tumor site as calculated using the CIBERSORT algorithm. Color of the dots indicate positive (blue) or negative correlations (red). The size of the dots is proportional to the Spearman Rho value. **e** Unsupervised clustering of 1318 Cluster B samples from 15 cohorts according to the gene set signature scores using the correlation linkage and ward.D method allows to separate the samples in Cluster B with an EMT phenotype: Cluster B1 (green) or proliferative phenotype: Cluster B2 (orange). PAM50 subtypes and ER status are annotated on the top of the heatmap. **f**, **g** Kaplan–Meier survival curves for Cluster B1 (green) and Cluster B2 (orange). In all METABRIC (**b**) and TCGA (**c**) samples. $p$ Values are from log-rank tests. Kaplan–Meier display breast cancer-specific survival for the METABRIC and relapse-free survival for the TCGA.

Cluster A and Cluster C separately (Bonferroni-corrected $p$ value < 0.0001; Supplementary Data 3). These genes were associated with stem cell biology and EMT, as shown by the gene set enrichment analysis (GSEA) using the H and C2 collection of the MsigDB[31] (Fig. 6a).

To further characterize the relationship between the immune clusters and cancer cell phenotype, we used gene sets associated with EMT, stem cells, hypoxia, and proliferation. In total, 11 gene sets from the MsigDB and an additional EMT-related signature from Tan et al.[32] were selected (Supplementary Data 3). We

calculated per cluster and cohort an average gene set enrichment score using the GSVA method; this score reflects the activity of each pathway/gene set in an immune cluster[33]. Unsupervised clustering of averaged-gene-set scores clearly separated the immune Clusters A and C, while Cluster B was divided into two subgroups (Fig. 6b). These results suggested an association between immune clusters and the stem cell/EMT-related gene signatures.

**Two mutually exclusive phenotypes in breast cancer**. Through unsupervised clustering of GSVA enrichment scores, we identified two mutually exclusive gene signatures in breast cancer, (i) one associated with proliferation and embryonic stem cell-like phenotype and (ii) and the other with EMT and mammary stem cell phenotype.

A proliferative phenotype was dominating Cluster C (Supplementary Fig. 11A), the same was observed when gene set scores were calculated for each METABRIC sample (Supplementary Fig. 11B). In Cluster B, the average gene set scores were either high for EMT or proliferation-related signatures (Supplementary Fig. 11C). At the sample level in the METABRIC, we observed a similar pattern with samples having the one or the other state activated (Supplementary Fig. 11D). Cluster A showed low scores for both the EMT and proliferative states (Supplementary Fig. 11E, F).

To formally identify which gene set scores explained Cluster B, we tested how each gene set contribute to Cluster B vs Clusters A and C using generalized linear models. EMT signatures contributed positively to Cluster B while proliferation and cell motility were associated with Clusters A and C (Fig. 6c). We also tested which gene set score explained Cluster B when compared separately to Cluster A (Supplementary Fig. 12A) or Cluster C (Supplementary Fig. 12A). We found in both cases EMT scores being a significant explanatory variable of Cluster B. However, EMT signature scores alone were not of strong prognostic value according to Cox regression analysis (Supplementary Table 9). Overall, these results suggest a mutually exclusivity between EMT and proliferation in breast cancers. They also suggest that only when accompanied by a certain immune contexture the EMT or the proliferative phenotype result in poor prognosis.

**Correlation between tumor phenotype and immune infiltration**. As immune clusters were associated with both (i) immune cell types and (ii) gene set signatures, we formally assessed the relation between immune infiltration (CIBERSORT) and cancer cell characteristics (gene set scores). Figure 6d shows that the proliferation and EMT scores correlate significantly with different type of immune cells. Notably, high EMT scores are associated with macrophages M2, resting mast cells, and resting memory T cells while high proliferation is correlated with a more active adaptive tumor microenvironment (macrophages M1, T helper cells, activated dendritic cells, and active memory T cells). These data suggest a continuum between the cancer cell phenotype and the composition of the tumor microenvironment.

**Heterogeneity in gene set scores within Cluster B**. Cluster B was dominated by samples with pro-tumorigenic immune infiltration and high EMT signal; however, ~35% of Cluster B samples also exhibited a proliferative phenotype. To explore this heterogeneity within Cluster B, we grouped samples according to the gene signature scores in an unsupervised manner into B1 dominated by the EMT phenotype and B2 by the proliferation (Fig. 6e).

In the METABRIC and TCGA, B2 cases with the proliferative phenotype had a worse outcome (Fig. 6f, g, also see Supplementary Fig. 13 in which survival probabilities of B1 and B2 are plotted with Cluster A and Cluster C). While we were able to

identify a difference in survival between Cluster B1 and B2 in METABRIC and TCGA, for other smaller cohorts, it was difficult to conclude, as further splitting Cluster B resulted in small groups. To further assess whether the heterogeneity in gene set scores was accompanied by heterogeneity in immune contexture, we sought for differences in specific immune cell types between sub-clusters B1 and B2. Unsupervised clustering in Supplementary Fig. 14 showed that the two sub-clusters B1 and B2 both have a pro-tumorigenic/resting immune microenvironment.

Altogether, the two mutually exclusive states within Cluster B may be relevant in regard to prognosis; however; a unifying factor of Cluster B is the presence of a pro-tumorigenic/resting immune microenvironment.

## Discussion

The tumor microenvironment plays an important role in breast cancer pathogenesis. We provide a new immune-related subtype in breast cancer with relevance for prognosis and response to neoadjuvant chemotherapy in both ER-positive and ER-negative cases. The herein described immune clusters are dependent on both the abundance and composition of the immune infiltrate and are independent of other prognostic factors, including PAM50.

Through unsupervised clustering using the expression of genes part of the nCounter® PanCancer Immune Profiling Panel, we identified in FFPE and fresh frozen breast tumors, three clusters of patients. These clusters were (i) associated with total levels of immune infiltration and with specific immune microenvironment, (ii) provided an independent prognostic information, and (iii) revealed two mutually exclusive breast cancer phenotypes.

As the immune clusters provided an independent prognostic value in breast cancer, we developed a simple method that refined and accurately predicted whether a sample falls in the poor prognosis cluster (Cluster B) or not. We tested our method successfully in 15 cohorts, spanning 6101 breast cancer samples. We demonstrate using different and complementary statistical approaches the strength of the immune cluster as a new prognostic biomarker.

Through phenotypical characterization of the immune clusters, we also identified two mutually exclusive states in breast cancers, one associated with EMT and the other with proliferation. A similar observation of two mutually exclusive states: proliferative and EMT, was recently reported in a pan-cancer genomic analysis of metastatic tumors[34]. Our study therefore suggests that such a mutual exclusion could be extended to primary breast tumors and possibly to other primary cancer types. The EMT process has often been associated with metastasis[35]; it has been also previously suggested that transcription factors such as TWIST1, which may drive the EMT process, need to be turned off for the cancer cell to proliferate[36]. Such a mechanism may explain why these two processes could not coexist in cancer cells.

Samples with the EMT or proliferative phenotype were found in the poor prognosis cluster (Cluster B). About 65% of the samples in Cluster B had an EMT-like phenotype. We further found that this dominating phenotype could help explain Cluster B when compared to Clusters A and C using generalized linear models. As opposed to that, 35% of the samples in Cluster B had a proliferative phenotype like most of the Cluster C samples. Proliferation in Cluster C associated with infiltration of active anti-tumorigenic immune cells, while proliferative samples in Cluster B had infiltration of immune cells less likely to eradicate cancer cells (macrophages M2, resting mast cells). This indicates that in breast cancer a proliferative phenotype associated with a non-adapted, pro-tumorigenic, resting immune microenvironment relates to an adverse outcome as indicated by the Kaplan–Meier analysis (Fig. 6f, g).

Many studies have suggested that EMT drives an aggressive tumor phenotype in breast cancer[37,38]. However, recent studies have questioned the role of EMT in tumorigenesis, progression, and metastasis[39]. Importantly, we show here that specific immune infiltration is associated with the EMT process in breast cancer. As a recent study also suggests[40], we highlight that immune contexture is an important factor to consider when evaluating the role of EMT during cancer pathogenesis.

Using CIBERSORT[19] to infer for specific immune cell infiltration, we found the EMT state highly correlated with infiltration of resting mast cells, macrophages M2, natural killer (NK) cells, and resting memory T cells. It has been previously suggested that EMT could be associated with a pro-tumorigenic microenvironment in lung cancer[41]. In esophageal squamous cell carcinoma, M2 macrophages promote migration, invasion, and enhance EMT[42]. On the other hand, mast cells have been associated with angiogenesis in breast cancer[43]. Based on several gene expression datasets, our current results demonstrate that the EMT process is accompanied with infiltration of pro-tumorigenic/resting immune cell types. The presence of antitumorigenic immune cells, like NK cells, has also been found to be highly correlated with the EMT in melanoma[44].

In Cluster C, a proliferative phenotype was found to be correlated with infiltration of activated dendritic cells, T helper cells, macrophages M1, and CD4 memory T cells. These cell types reflect an antitumoral microenvironment. Cluster C is dominated by both a highly proliferative phenotype and high infiltration of antitumoral cell types. One may argue that chemotherapies may successfully eradicate such proliferative tumors with the support of an antitumoral microenvironment; explaining a better outcome of these patients and the higher rate of responders to neoadjuvant chemotherapy in Cluster C (Fig. 4e).

Low immune infiltration in Cluster A associated with neither the proliferative nor the EMT state, which may indicate a less aggressive tumor phenotype.

Previous studies have suggested that basal-like breast cancers display a high metastatic ability associated with mesenchymal features[45]. Sarrio et al.[46] further showed that several markers of EMT were upregulated in basal-like breast cancers[46]. Our study shows using recent algorithms that the EMT phenotype is enriched in Cluster B. In breast cancer, a recent gene transcriptional profiling has identified an EMT gene expression signature associated with claudin-low and metaplastic breast cancers[47]. However, the claudin-low subtype in the METABRIC cohort did not correlate with Cluster B.

Our study suggests that targeting the primary pathways involved in EMT such as transforming growth factor-b[48], E-Cadherin[49], WNT/B-catenin pathway[50], Notch[51], hypoxia, or tumor necrosis factor-alpha[52] are interesting opportunities for therapeutic intervention for patients with the worse prognosis (Cluster B). More importantly, the macrophage re-education strategy, which proposes to remodel M2 type of macrophages into an anti-tumor, "M1-like" mode[53], could be beneficial for Cluster B patients.

In the era of modern immunotherapy, a few clinical trials using immune checkpoint inhibitors have been conducted in breast cancers and have been planned to be combined with immunogenic chemotherapy or radiation therapy. The results of the first clinical trials using monoclonal antibodies against immune checkpoint inhibitors have recently been communicated and show some degree of response especially in certain subpopulations[54,55]. Our study suggests that considering both the immune cell types infiltrating the tumor and the main state of the tumor (EMT or proliferative) will precise treatment decisions and improve response to these new treatment strategies.

## Methods

**Gene expression analysis from FFPE**. Operable early breast cancer patients were included in the Oslo1 micrometastasis observational study between 1995 and 1998[56]. Informed consent has been obtained from all participants and the study was approved by the local ethical committee (S-97103). FFPE were collected for a subset of patient that also had fresh primary tumors collected for detailed molecular analyses, a cohort called MicMa. Only patients within the MicMa ($n = 96$) subset were included in the current analysis. FFPE tissue was first examined with H&E staining to determine the tumor area and dissection was performed to mainly include tumor tissue. RNA purification was performed using the Roche® High Pure FFPET RNA Isolation Kit; ≥1–5 10-µm FFPE slides were used for each tumor. A minimum of ~100 ng of total RNA was used on the nCounter platform (Nanostring Technologies, Seattle, WA, US) and the PanCancer Immune Profiling Panel[57]. Data were normalized using all housekeeping genes and log base 2 transformed.

**RNA-seq analysis of the OSLO2-EMIT0 cohort**. The OSL2 breast cancer cohort is a study collecting material from breast cancer patients with primary operable disease in several hospitals in south-eastern Norway. Inclusion of patients started in 2006 and is still ongoing. The study was approved by the Norwegian Regional Committee for Medical Research Ethics (approval number 1.2006.1607, amendment 1.2007.1125). Patients gave written consent for the use of material for research purposes. All experimental methods performed are in compliance with the Helsinki Declaration. Tumor tissue was cut into pieces and mixed before distribution to RNA extraction. RNA was isolated using the QIAgene kit Allprep DNA/RNA/miRNA universal on the QIAcube machine and method (Qiagen). Quality control was performed by Nanodrop ND-1000 (NanoDrop Technologies) and BioAnalyzer 2100 (Agilent) analysis. All RNA had RNA Integrity Number (RIN) ≥ 6. We used Illuminas TruSeq Stranded mRNA Library Prep Kit for the automated NeoPrep Library Prep System (Illumina). Starting amount was 120 ng total RNA and we used Illuminas NextSeq500 sequencers (2 × 75 bp). Raw sequencing read data were demultiplexed and filtered using Bowtie2 against ribosomal, phiX174, and UCSC RepeatMasker sequences. The sequence data were processed as described previously.[58] Log-transformed FPKM RNA-seq gene expression data at GEO are available at GSE135298. Raw data are available at EGAS00001003631.

**Data collection and processing**. Publicly available expression data from breast cancer cohorts were used in this study. Patients' consents and ethical approval are available in the respective original articles the datasets were published with (Supplementary Table 2). Expression data were obtained from Gene Expression Omnibus, the European Genome-phenome Archive, ArrayExpress, or TCGA data portals. For survival analyses, we selected studies with >100 samples and relevant survival data from patients with invasive breast tumors sampled at the time of surgical resection without neoadjuvant treatment. Survival data were of four types: relapse-free survival, distant metastasis-free survival, OS, or breast cancer-specific survival.

For analysis of response to neoadjuvant chemotherapy, we selected cohorts of patients treated with a chemotherapeutic regimen and for which gene expression has been profiled from the primary tumor prior to treatment. pCR was assessed at the time of surgery at the end of treatment and refers to the total elimination of cancer cells at surgery.

Except for the METABRIC cohort for which the ER status has been extensively used and defined, we used gene expression data together with the R package *optim* to systematically infer for ER status using a two-component Gaussian finite mixture model using maximum likelihood estimation as previously described[59]. Classification into the PAM50 intrinsic molecular subtypes was performed based on gene expression data using the genefu package in R[3].

**Gene set enrichment analysis**. Gene set enrichment analysis was performed using the Molecular Signatures Database v4.0 (MSigDB[31]) H and C2 collections. Enrichment was assessed by hypergeometric testing.

**Unsupervised clustering to obtain immune clusters**. First a correlation matrix was calculated to assess the dependence between samples initially based on the expression of the 760 genes in the nCounter® PanCancer Immune Profiling Panel and later with the 509 genes that are present in all clustered datasets (training in Supplementary Table 2). Hierarchical clustering of patients' correlation matrix was performed using the R package pheatmap v1.0.12 using correlation as clustering distance and ward.D as linkage. Clusters were identified using the cutree function. To determine the optimal number of clusters for each cohort, we used the silhouette analysis of KMeans using the cluster R package; for most of the cohorts assessed, three clusters was a better pick than more numerous clusters.

**Nanodissect analysis, lymphoid and myeloid scores**. The algorithm Nanodissect (http://nano.princeton.edu) was used as previously described to predict for lymphoid and myeloid infiltration[24,25]. Breast collection data (May 2013), which

contains 17,940 genes measured on 622 arrays, was inspected for genes specifically expressed in lymphoid or myeloid cell types and not expressed in mammary gland or mammary epithelium. The genes with >65% probability to be positive lymphocyte- or myelocyte-specific standard genes as opposed to mammary gland or epithelium were used in downstream analysis. Nanodissect scores for lymphocyte or myelocyte infiltration reflect the average expression of the respective genes (Supplementary Data 4) in a sample.

**CIBERSORT analysis**. The algorithm CIBERSORT was used on normalized expression data to infer the absolute proportions of 22 types of infiltrating immune cells. CIBERSORT is a deconvolution algorithm that uses a set of reference gene expression values (547 genes) to predict 22 immune cell type proportions from bulk tumor sample expression data by using support vector regression[19]. To assess the reliability of the deconvolution method, CIBERSORT derives a $p$ value for each sample. CIBERSORT software package was obtained from the developers, and analysis was performed by using the default signature matrix at 1000 permutations.

**Single-sample GSEA (GSVA)**. Gene set analysis was carried out using the GSVA Bioconductor package v1.30.0[33]. We curated gene sets for various epithelial mesenchymal transition, stem cell, proliferation, and cell cycle-related pathways (Supplementary Data 3). For each sample, a score for the enrichment of a set of genes using gene expression profile was obtained.

**Binomial logistic regression to predict immune clusters**. We used binomial logistic regression through the glmnet v2.0–16R package[60] to develop a method that allows to assign any given sample to the group with the worse prognosis or not without resorting to unsupervised clustering. This predictor method is highly efficient for smaller cohorts and allow to assign class to single samples. To perform the analysis, we mean centered datasets and set up a logistic regression using the binomial distribution to predict categorical response of the two possible outcomes: being in the bad prognosis group or not. This approach gave a signature of target genes that together captured the variation associated with the two categories (Supplementary Data 1).

Patients were divided into Cluster B or Clusters A and C groups according to the following index for patient $i$:

$$\text{Index}_i = \sum_{g=1}^{n} \beta_g . X_{gi} \qquad (1)$$

where $g$ is the target (gene), $n$ is the number of targets, $\beta_g$ is the lasso coefficient for target gene and $X_{gi}$ is the gene expression value in sample $i$. If index for patient $i$ is higher than the intercept = 1.206538657, sample is assigned to Cluster B.

**Pathological assessment of immune infiltration**. Vascular invasion, inflammatory cell infiltrate, and necrosis, including relation of tumor cells/tumor stroma, were evaluated on slides stained with H&E as previously described[61]. Using a simple microscope, subjective categorization of inflammatory cell infiltrate into the categories of "low," "moderate," "high," and "severe" was performed based on the frequency of mononuclear inflammatory cell infiltration observed in the invasive tumor.

**ROR score calculation**. ROR scores for each sample were calculated as described in ref. [3], ROR-Score = 0.05 × Basal + 0.12 × Her2-enriched − 0.34 × Luminal A + 0.23 × Luminal B; where Basal, Her2-enriched, Luminal A, and Luminal B are the correlation of each sample to the centroid obtained using the genefu package in R.

**Statistical, survival, multivariable Cox regression analysis**. All analyses were performed in the R version 3.3.2. Unless otherwise stated, results were considered statistically significant, if $p$ value < 0.05. Kaplan–Meier estimator and log-rank tests were performed using the functions Surv, survfit, and survdiff (R package survival v2.42–3). Multivariable Cox regression analyses were used to test the independent prognostic value of the immune clusters using the R package *survival* and the coxph function. Mann–Whitney $U$ or Kruskal–Wallis tests were used to assess statistical significance within boxplots.

In the box-and-whisker plots, the line within each box represents the median. Upper and lower edges of each box represent 75th and 25th percentile, respectively. The whiskers represent the lowest datum still within [1.5 × (75th − 25th percentile)] of the lower quartile and the highest datum still within [1.5 × (75th − 25th percentile)] of the upper quartile.

To identify differentially expressed genes between clusters, we used a $t$ test followed by Bonferroni correction of the $p$ value. A strict corrected $p$ value ($p <$ 0.0001) was used to identify differentially expressed genes.

NRI and IDI were calculated using the survIDINRI v1.1–1 R package. To assess the 95% CI and $p$ values for the IDI and NRI, a standard bootstrap method was used with resampling performed 500 times. NRI and IDI were assessed at the maximum follow-up time as presented in the Kaplan–Meier survival analysis to assess the improvement in performance of the survival model.

Forest plots were obtained using the forestplot v1.7.2 R package and represent for the univariate and multivariate analysis the hazard ratio and their 95% CI. Boxes represent hazard ratios and are inversely proportional to the width of the CI, horizontal lines are 95% CI.

Correlation plot using the corrplot v0.84 package visualizes Spearman correlations, only False Discovery Rate-corrected significant correlation are visualized and colored according to directionality of the rho values. Size of the dots are proportional to the rho value.

**Reporting summary**. Further information on research design is available in the Nature Research Reporting Summary linked to this article.

## Data availability

All data used in this study are publicly available or can be downloaded through the European Genome-phenome Archive (EGA)—EMBL-EBI portal. We mainly used gene expression datasets from breast cancers summarized in this study in Supplementary Table 2. Data were downloaded with the study-specific normalization process. For initial clustering of the correlation matrix, no other normalization was performed. Further on, for binomial logistic regression and further downstream analysis (CIBERSORT, GSVA, differential expression) the datasets were mean centered. The source data underlying all Figures and Supplementary Figures are available in a source data file. The newly generated RNA-seq gene expression data for the breast cancer cohort OSLO2-EMIT0 is available at EGA with accession number EGAS00001003631. Log-transformed FPKM RNA-seq gene expression data at GEO are available at GSE135298. Newly generated, normalized log 2-transformed nCounter counts for the MicMa cohorts can be found in Supplementary Data 5.

## Code availability

To reproduce all figures published in this study, we provide all codes and relevant data in a source data file. In addition, the code to subtype the immune clusters are available online at http://eurostar.nebdal.no:5000/ as well as the codes to subtype using R or python are available at https://github.com/dnebdal/clusterscore.

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

## Acknowledgements

This study was supported by funding from the KG Jebsen Centre for Breast Cancer Research (SKGJ-MED-004) and the South Eastern Norway Health Authority (grant 2011042 to V.N.K.). Expression profiling was performed with funding from the Research Council of Norway (grant 193387/H10 to Anne-Lise Børresen-Dale and V.N.K.). X.T. is a postdoc fellow funded by the Norwegian Cancer Society (grant no. 419616111190). RNA-sequencing of frozen tumor samples was performed in the SCAN-B laboratory at Lund University, supported by grants from the Mats Paulsson Foundation and Mrs Berta Kamprad Foundation (2012/3657).

## Author contributions

X.T.: designed the study, performed analysis, wrote the manuscript. T.L.: performed analysis. D.N.: developed the webservice. A.H.R., T.S. and H.G.R.: provided critical points of view. E.B.: scoring of FFPE for inflammation H.O.O. and K.K.S.: provided tissue samples. J.A.K.: provided critical points of view. J.V.-C.: bioinformatic analysis. M.F. and E.U.D.: prepared samples for RNA-seq. L.G.S. and M.A.T.S.: prepared samples for nCounter analysis. A.F.: counseling on statistical analysis. O.G.: pathological inspection of FFPE samples. B.N.: designed the study, provided tissue samples. V.N.K.: designed the study, wrote the manuscript, supervised all steps of the study.

## Competing interests

The authors declare no competing interests.

## Additional information

## OSBREAC

Anne-Lise Børresen-Dale[1], Ellen Schlichting[11], Torill Sauer[12,13], Jürgen Geisler[9,14,15], Solveig Hofvind[16,17], Tone F. Bathen[18], Olav Engebråten[3,9,19], Gry Aarum Geitvik[1], Anita Langerød[1], Rolf Kåresen[9,20], Gunhild Mari Mælandsmo[19,21], Ole Christian Lingjærde[22,23], Helle Kristine Skjerven[24], Daehoon Park[25] & Britt Fritzman[26]

[11]Section for Breast and Endocrine Surgery, Oslo University Hospital, Ullevål, Oslo, Norway. [12]Department of Pathology, Akershus University Hospital, Lørenskog, Norway. [13]Institute of Clinical Medicine, Faculty of Medicine, University of Oslo, Oslo, Norway. [14]Department of Oncology, Akershus University Hospital, Lørenskog, Norway. [15]Division of Medicine, Akershus University Hospital, Lørenskog, Norway. [16]Cancer Registry of Norway, Oslo, Norway. [17]Oslo and Akershus University College of Applied Sciences, Faculty of Health Science, Oslo, Norway. [18]Department of Circulation and Medical Imaging, Norwegian University of Science and Technology (NTNU), Trondheim, Norway. [19]Department of Tumor Biology, Institute for Cancer Research, Oslo University Hospital, Oslo, Norway. [20]Department of Breast and Endocrine Surgery, Division of Surgery, Cancer and Transplantation, Oslo University Hospital, Oslo, Norway. [21]Department of Pharmacy, Faculty of Health Sciences, University of Tromsø, Tromsø, Norway. [22]Centre for Cancer Biomedicine, University of Oslo, Oslo, Norway. [23]Department of Computer Science, University of Oslo, Oslo, Norway. [24]Breast and Endocrine Surgery, Department of Breast and Endocrine Surgery, Vestre Viken Hospital Trust, Drammen, Norway. [25]Department of Pathology, Vestre Viken Hospital Trust, Drammen, Norway. [26]Østfold Hospital, Østfold, Norway

