## [Peer Review File · Nature Communications]

Reviewers' comments:

Reviewer #1 (Remarks to the Author):

In this manuscript, Tekpli and colleagues identify and characterize three immune-related gene expression subtypes in breast tumors. One such cluster, denoted "Cluster B," was found to associate with intermediate levels of immune content and adverse outcomes, and was found to consist of a uniform representation of PAM50 subtypes. Cluster B was also found to be prognostic within selected PAM50 subtypes, including basal breast cancer, and was prognostic independent of ER status and other major clinical indices. Gene set enrichment analyses revealed an enrichment of EMT-related processes in Cluster B along with reciprocal expression patterns for canonical stem cell/proliferation-related genes and mammary stem cell/EMT-related genes. Distinct correlations were identified between inferred leukocyte composition and two tumor states, as defined by contrasting expression patterns of previously defined EMT-related genes or proliferation-related genes.

Although the identification of novel breast cancer prognostic signatures would be beneficial if they can significantly improve upon existing clinical and molecular risk factors and/or yield new insights into breast cancer biology, the current manuscript falls short on several levels. For example, many groups have demonstrated associations between breast cancer clinical outcomes and immune content, EMT, proliferation, and/or stemness. The molecular signature of Cluster B may distinguish this study from others, but major questions remain about the biological interpretation and novelty of the findings, and whether they are clinically important (below). Strengths of the study include the generation of new breast tumor gene expression profiles and a single-sample classifier to predict the cluster membership of new samples.

Major comments:

1. As Cluster B appears to be heavily influenced by EMT, the novelty and added value of Cluster B is unclear. How does the expression of EMT-related genes compare to Cluster B in univariate and multivariate survival analyses? How do the authors reconcile their results with a recent study, in which the role of EMT-associated genes in mammary gland regeneration and breast tumorigenesis was called into question (PMID 29162812)? Separately, as a heterogeneous cluster mostly associated with EMT but also associated with a mutually exclusive proliferation signature, the authors should better explain how Cluster B is biologically meaningful.
2. The authors should assess whether the addition of Cluster B truly improves upon existing clinical and molecular risk factors using net reclassification improvement (NRI) and integrated discrimination improvement (IDI) indices.
3. While the authors attempted to characterize links between immune composition and breast tumor transcriptional heterogeneity, rather than focus on immunological distinctions between Clusters A, B, and C, they focused on differences between two gene sets, HALLMARK_EMT and BENPORATH_PROLIFERATION. The omission of the former is perplexing, seeing as the clusters were defined by immune-related genes and reflect overall immune content.
4. The authors appear to be over-interpreting the relationship(s) between previously described stem cell-associated gene sets and the presence of cancer stem cells (CSCs). CSCs are generally defined as self-renewing cancer cells that can differentiate into all of the cells of the neoplasm. Whether CSCs are sufficiently abundant to be detectable in bulk tumor expression profiles is a matter of active debate, and CSCs in breast cancer remain poorly defined. The main stem cell-related gene sets utilized by the authors are surrogates for actively dividing cells (WONG_EMBRYONIC_STEM_CELL_CORE) and basal epithelium (LIM_MAMMARY_STEM_CELL_UP). While the latter is associated with stem cell-like genes, it was derived from a comparative analysis of basal epithelium versus luminal epithelial cell subpopulations. Moreover, while cluster B is defined by both EMT and mammary stem cell-related genes, the authors of the Lim signature note that "the expression of these genes in tumor cells may reflect basal epithelial cell characteristics and not necessarily cells that have undergone an EMT" (http://software.broadinstitute.org/gsea/msigdb/cards/LIM_MAMMARY_STEM_CELL_UP). Finally,

while basal cells may harbor CSCs, the molecular signatures of such CSCs are largely unknown.

5. The ssGSEA results in Supplementary Figure 9 include a number of cell types that are clearly not present in the analyzed bulk tumor samples, yet are inferred to be present and correlated (in some cases highly) with HALLMARK_EMT by the expression of marker genes (e.g., astrocytes, mesangial cells, hepatocytes, and many others). These nonsensical results cast doubt on the validity of this analysis.

6. The authors identify three immune-related clusters in breast tumor FFPE samples using genes that are covered by the NanoString PanCancer Immune Profiling platform. Similar results were seemingly obtained by applying hierarchical clustering to the METABRIC cohort (microarrays) after subsetting on NanoString genes. The authors claim to have identified the same clusters in both cohorts, but no assessment of overlap in cluster assignments between the cohorts is presented. The logistic regression framework presented in Fig. S5 could be applicable here.

7. The authors state that they “provide evidence that not the amount per se but rather specific immune cell type mixtures in the tumor microenvironment associate with poor prognosis in breast cancer.” This conclusion appears to be odds with the presented results, which clearly show that patients with “intermediate” levels of total immune content are associated with worse outcomes. Moreover, whether Cluster B harbors a distinct immune repertoire, as compared to Clusters A and C, has not been shown. If the amount of immune content is indeed irrelevant, then the authors should demonstrate this with a multivariate model that also considers Cluster B as a covariate.

8. Is Cluster B predictive of response to chemotherapy? There are publicly available datasets that can be leveraged to help address this.

9. The single-sample classifier should be made publicly available as source code and ideally as a web service.

Minor comments:

1. There are numerous grammatical errors and typos that affect readability.
2. Figure 1 caption: the reference to “(A & C)” should be “(A & D)”.

Reviewer #2 (Remarks to the Author):

In this manuscript, Tekpli et. al. used patient tumor specimens and performed expression array of genes that are related to immune infiltration. They identified three clusters A, B, C, that show lymphocyte infiltration from low to high. They found that these three clusters exist in several sources of tumor samples including those from TCGA and METABRIC. They further showed that patients fall into Cluster B, which have intermediate lymphocyte infiltration, are associated with poor survival compared to Cluster A and C. The authors used a binomial logistic regression penalized by the lasso method and obtained a set of genes that was able to determine whether the patients belong to Cluster B. However, the false positive rate was $\sim 30\%$ ($=1 - 68.8\%$). The authors also found that immune clusters are an independent prognostic factor in breast cancer, while the gene profiles of cluster B do not predict well on risk of recurrence (ROR). They further performed gene enrichment analysis and showed connections between immune Clusters and EMT signatures. However, the claim that “these results identify an EMT-stem-cell-like phenotype in Cluster B which associate with poor prognosis.” does not hold from the data presented. Essentially, Cluster A and Cluster C tumors from different collection sources showed more uniformed gene signatures, whereas Cluster B showed varied gene signatures among different tumor cohorts. In other words, enriched gene signatures in Cluster B are not consistent among different cohorts. Collectively, the reviewer finds that the study showed some interesting findings. However, the results are still at an observational stage and some of the conclusions are not justified.

Reviewers' comments:

Reviewer #1 (Remarks to the Author):

In this manuscript, Tekpli and colleagues identify and characterize three immune-related gene expression subtypes in breast tumors. One such cluster, denoted "Cluster B," was found to associate with intermediate levels of immune content and adverse outcomes, and was found to consist of a uniform representation of PAM50 subtypes. Cluster B was also found to be prognostic within selected PAM50 subtypes, including basal breast cancer, and was prognostic independent of ER status and other major clinical indices. Gene set enrichment analyses revealed an enrichment of EMT-related processes in Cluster B along with reciprocal expression patterns for canonical stem cell/proliferation-related genes and mammary stem cell/EMT-related genes. Distinct correlations were identified between inferred leukocyte composition and two tumor states, as defined by contrasting expression patterns of previously defined EMT-related genes or proliferation-related genes.

We thank the reviewer for the careful evaluation of our study and the many positive and constructive comments/suggestions we received. The additional analyses required strengthened our conclusions. We answer to all the questions / comments below. Major changes are highlighted in the revised manuscript.

Although the identification of novel breast cancer prognostic signatures would be beneficial if they can significantly improve upon existing clinical and molecular risk factors and/or yield new insights into breast cancer biology, the current manuscript falls short on several levels. For example, many groups have demonstrated associations between breast cancer clinical outcomes and immune content, EMT, proliferation, and/or stemness. The molecular signature of Cluster B may distinguish this study from others, but major questions remain about the biological interpretation and novelty of the findings, and whether they are clinically important (below). Strengths of the study include the generation of new breast tumor gene expression profiles and a single-sample classifier to predict the cluster membership of new samples.

We are grateful to the reviewer for helping us formulate the novelty of our study and for acknowledging that it could be clinically relevant and/or bring new insights into breast cancer biology.

The major novelties are that we identify one cluster (of three), which consists of all of the established PAM50 subtypes and yet has the worse prognosis. We then formally prove that the prognostic value of Cluster B is independent of and adds to that of PAM50, which has dominated the breast cancer field in the recent years. This may both explain heterogeneity within the PAM50 subclasses and open for new therapeutic strategies to target the immune component.

The next important message of our paper is the discovery that despite the common immunological background in Cluster B (which allowed us to discover it in the first place), the patients in this cluster present with tumors of 2 distinct phenotypes, EMT-like and Proliferative.

We agree that clinical outcomes in Breast Cancer have been associated EMT, proliferation, and/or stemness previously; however, no such strong immune-related prognosis marker has been described before. In addition, we discover an interplay between the immune component and EMT/proliferation which, to our knowledge, has never been described. We hope we can now convince on the novelty of these clusters, to back our claims, we:

- demonstrate the strength and clinical value of the immune clusters using complementary statistical analyses including Cox multivariable regression analysis with AIC index and NRI / IDI indices (Table 1, Supplementary Table 4, 5, 7 & 8).
- specify the immune cell types found in the immune clusters. Many studies report the clinical importance of high and low infiltration without considering the composition. Here we show that the content/quality of immune cells is as important as their quantity as we identify a cluster with "intermediate" immune infiltration and yet worse survival (Figure 5)
- assess the immune clusters according to response to neoadjuvant chemotherapy to show that patients in Cluster B, have a poor response to treatment (Figure 4E) while Cluster C contains 60% of the responders
- characterize the correlation between specific immune infiltration and cancer cell phenotype (Figure 6D)
- validate our findings in three additional cohorts (Figure 3 and Supplementary Figure 5).

Below are our specific answers to all questions and description of other important changes.

Major comments:

1. As Cluster B appears to be heavily influenced by EMT, the novelty and added value of Cluster B is unclear. How does the expression of EMT-related genes compare to Cluster B in univariate and multivariate survival analyses?

In the previous version of the manuscript we did not assess the specific immune cell types associated with Cluster B, we now show that a specific immune contexture characterizes Cluster B, which we think is the main novel finding.

Cluster B is strongly associated with EMT as described (p11, last paragraph and Figure 7). We also show through generalized linear models that the EMT-gene scores help explaining Cluster B phenotype (Figure 6C).

EMT is however not sufficient to explain the strong novel prognostic values carried by Cluster B. Indeed, when assessing the EMT-gene-signature scores in uni- and multivariate analysis, the EMT scores alone are poorly associated to survival (Supplementary Table 8 and p 11, first paragraph). The novelty and the added value of our findings are the association of this EMT signature with specific types of immune cells.

Therefore, the immune contexture discovered here to define Cluster B captures more than EMT signal alone.

How do the authors reconcile their results with a recent study, in which the role of EMT-associated genes in mammary gland regeneration and breast tumorigenesis was called into question (PMID 29162812)?

The study pointed by the reviewer investigated the role of EMT mainly in triple negative breast cancers, authors observed no or poor association between EMT-associated genes and mammary gland development as well as breast tumorigenesis.

We, ourselves, show through the univariate analysis that EMT scores alone do not carry strong prognostic value (Supplementary Table 8). Our immune clusters, and especially Cluster B, span across all the breast cancer subtypes. We believe that an important feature captured by the immune clusters is the association of an aggressive tumor phenotype (EMT OR proliferation) with a specific immune microenvironment which drives the poor outcome.

Such an interaction between the cancer cells and their microenvironment was perhaps not recapitulated in the experimental system of PMID 29162812. We comment this specific study and topic in our discussion (p13,3rd paragraph).

Separately, as a heterogeneous cluster mostly associated with EMT but also associated with a mutually exclusive proliferation signature, the authors should better explain how Cluster B is biologically meaningful.

The main finding from our work is that we identify immune clusters which are a new independent prognostic factor in breast cancer and these clusters are constituted by specific immune cell types, we think, this is an important point defining biologically the clusters. What kind of tumor cells and tumor related signatures would the immune cluster be associated with? We had no hypothesis for before performing the analyses. The fact that Cluster B captures two mutually exclusive signatures was interesting to us and is worth reporting. We have in the new version of the manuscript highlighted and confirmed through new analysis that:

- (i) Cluster B has a medium degree of infiltration quantitatively ('old' Figure 1B and 1E),
- (ii) this intermediate type of infiltration is mainly pro-tumorigenic and not adapted to tumor eradication (Macrophages M2, resting Mast cells, resting memory T cells) (new Figure 5).

We dissect in more details the composition of Cluster B in regard to the mutually exclusive proliferation and EMT signature by identifying two sub-clusters (B1 and B2) and their association with prognosis (new Figure 7).

Despite the dichotomy in Cluster B regarding EMT vs proliferation, unifying factors are (i) medium type of immune infiltration with specific immune cell types which are pro-tumorigenic, (ii) strong association with poor outcome across 15 breast cancer cohorts, (iii) association with response to neoadjuvant chemotherapy.

2. The authors should assess whether the addition of Cluster B truly improves upon existing clinical and molecular risk factors using net reclassification improvement (NRI) and integrated discrimination improvement (IDI) indices.

We thank the reviewer for this suggestion which led us to apply additional statistical tests to assess the added value of the immune clusters upon other existing clinicopathological and molecular features.

As suggested, we used NRI and IDI against the ROR scores, which are recapitulating the molecular subtypes (PAM50) (PMID: 29137653). The results are presented in Supplementary Table 5 and first half of p8.

For 11 cohorts, with relevant survival data we invariably find positive NRI and IDI clearly showing that immune clusters add some value in classifying breast cancer patients according to prognosis. Bootstrapping for confidence interval construction for NRI and IDI showed that at least for several cohorts the immune clusters were adding value to the ROR scores (p value < 0.05) which demonstrates the usefulness of the immune clusters and their clinical importance.

We also conducted multivariable Cox regression analysis which demonstrates that in all cohorts the immune clusters are a necessary and important factor to model and explain breast cancer survival as

demonstrated by the AIC indexes and the backward selection of variables (Table 1 and Supplementary Table 4).

3. While the authors attempted to characterize links between immune composition and breast tumor transcriptional heterogeneity, rather than focus on immunological distinctions between Clusters A, B, and C, they focused on differences between two gene sets, HALLMARK_EMT and BENPORATH_PROLIFERATION. The omission of the former is perplexing, seeing as the clusters were defined by immune-related genes and reflect overall immune content.

We are very grateful for this comment, which led us to closely characterize the contexture of the immune clusters using CIBERSORT which infers for the presence of 22 immune cell types in the tumor microenvironment. This led us to novel findings, i.e. the identification of the cell types 'overrepresented' in Cluster B when compared to Cluster A & C.

In Cluster B, we identified pro-tumorigenic cell types such as resting mast cells, M2 type of Macrophages as well as resting/inactive immune cells (Figure 5A and 5B, second half of p9). Using generalized linear models, we also show that these pro-tumorigenic cell types are characteristic of Cluster B in comparison to Cluster A&C (Figure 5C, second half of p9).

These analyses shed new light on the immune microenvironment associated with Cluster B and appeared as an important factor unifying Cluster B even across sub-clusters B1 and B2 (Supplementary Figure 10).

4. The authors appear to be over-interpreting the relationship(s) between previously described stem cell-associated gene sets and the presence of cancer stem cells (CSCs). CSCs are generally defined as self-renewing cancer cells that can differentiate into all of the cells of the neoplasm. Whether CSCs are sufficiently abundant to be detectable in bulk tumor expression profiles is a matter of active debate, and CSCs in breast cancer remain poorly defined. *The main stem cell-related gene sets utilized by the authors are surrogates for actively dividing cells (WONG_EMBRYONIC_STEM_CELL_CORE) and basal epithelium (LIM_MAMMARY_STEM_CELL_UP). While the latter is associated with stem cell-like genes, it was derived from a comparative analysis of basal epithelium versus luminal epithelial cell subpopulations. Moreover, while cluster B is defined by both EMT and mammary stem cell-related genes, the authors of the Lim signature note that "the expression of these genes in tumor cells may reflect basal epithelial cell characteristics and not necessarily cells that have undergone an EMT" (http://software.broadinstitute.org/gsea/msigdb/cards/LIM_MAMMARY_STEM_CELL_UP). Finally, while basal cells may harbor CSCs, the molecular signatures of such CSCs are largely unknown.*

We agree with the reviewer. The gene sets related to stem cells are not necessarily reflecting the presence of cancer stem cells in the tumor and rather a proliferative and / or more plastic tumor phenotype. It is not our wish to over interpret or interpret wrongly the meaning of these signatures, which remain in our view very relevant as they are part of the mutual exclusion in Cluster B. We have therefore not used the term cancer stem cell in the manuscript and modified our interpretation of the stem cell - related gene sets.

We are now using the gene set scores to (i) approach the phenotypes associated with Cluster B (Figure 6A and 6C), (ii) define the two mutually exclusive phenotypes in breast cancer (EMT vs proliferation)

(Figure 6B) (iii) identify specific immune cells correlated with the two mutually exclusive states (Figure 6D), (iv) split Cluster B into B1 and B2 (Figure 7A).

5. The ssGSEA results in Supplementary Figure 9 include a number of cell types that are clearly not present in the analyzed bulk tumor samples, yet are inferred to be present and correlated (in some cases highly) with HALLMARK_EMT by the expression of marker genes (e.g., astrocytes, mesangial cells, hepatocytes, and many others). These nonsensical results cast doubt on the validity of this analysis.

We agree with the reviewer that the interpretation of the xCell / ssGSEA output is complicated and should be taken with caution. Originally, this supplementary Figure was supporting the result of CIBERSORT.

However, some cell types not present in breast tissue are also included in the output of the xCell analysis. We initially interpreted this as a byproduct of a more plastic tumor phenotype, as it has been shown that breast tumors can express neuronal (PMID:24996968), melanocyte (PMID:19699574) or hepatic (PMID:25351134) genes.

Even though the tumor may express a wide range of marker genes of different cell types, we still agree with the reviewer that the xCell algorithm should not indicate that such cells are present in the tumor microenvironment. We therefore calculated a p value for each xCell score with the null hypothesis that the cell type is not present in the mixture as suggested by the authors (Aran et al. 2017). However, as these p -values were not helping to filter out cell types not present in breast tumors, we were concerned by the relevance of the xCell results and decided to remove this analysis from the study.

6. The authors identify three immune-related clusters in breast tumor FFPE samples using genes that are covered by the NanoString PanCancer Immune Profiling platform. Similar results were seemingly obtained by applying hierarchical clustering to the METABRIC cohort (microarrays) after subsetting on NanoString genes. The authors claim to have identified the same clusters in both cohorts, but no assessment of overlap in cluster assignments between the cohorts is presented. The logistic regression framework presented in Fig. S5 could be applicable here.

We are not sure if we interpret the reviewer's comment/question correctly. FigS5 (from the previous submitted version) presented the ROC curve showing the success of the prediction of the cluster A vs C by binomial logistic regression, this led to an AUC=95.5 when considering all cohorts. For the Metabric and the Micma separately, the AUC were perfect (AUC=100).

We understood the question as: 'do we obtain the same clusters when we use a subset of the original 760 genes and / or change platform to measure the gene expression?' To hopefully answer it, we repeated the clustering on a subset of samples on which gene expression has been measured on two different platforms. Clustering was performed (i) with 95 samples from the MicMa with 760 genes measured by Nanostring in FFPE and (ii) with 104 samples from the same cohort with 509 of the 760 genes which were measured by Agilent technology. Note that the 509 genes are the overlap of 'probes' found in all microarrays and RNA-seq data used in the study

For the 79 samples for which gene expression was measured with the two methods, we show a high degree of concordance in cluster assignment using either the 760 or 509 genes: Fisher exact test <

0.0001. We hope that this experimental, rather than statistical validation will be satisfactory and answer the question, the way we understood it.

To summarize, we obtain similar clusters even when (i) changing the platform for measuring gene expression and (ii) using only a subset of the gene list for clustering. This is now indicated in the manuscript (p 4, second paragraph of the results).

7. The authors state that they “provide evidence that not the amount per se but rather specific immune cell type mixtures in the tumor microenvironment associate with poor prognosis in breast cancer.” This conclusion appears to be odd with the presented results, which clearly show that patients with “intermediate” levels of total immune content are associated with worse outcomes. Moreover, whether Cluster B harbors a distinct immune repertoire, as compared to Clusters A and C, has not been shown. If the amount of immune content is indeed irrelevant, then the authors should demonstrate this with a multivariate model that also considers Cluster B as a covariate.

If only amount and not quality of immune cells would be associated with poor outcome, then one would expect to see a "dose-response" relationship between amount of infiltration and response, i.e. the more infiltration, the best response, or the least infiltration- the poorer response. When we saw that the worse prognosis was not in any of the extreme groups but in the intermediate one, we concluded that it is perhaps not only amount that matters. This is why we formulated the sentence in question, albeit not fully supported by additional evidence. We thank the reviewer for pointing that out. We believe we have now performed the necessary analysis to support this statement and it became an important part of our results.

First, we demonstrate using multivariable Cox regression analysis that the total lymphocyte infiltration is poorly associated with survival (Supplementary Table 6). We therefore went on to characterize the immune cell types enriched in each cluster.

Second, through unsupervised clustering of the inferred infiltration of 22 immune cell types, we found pro-tumorigenic and resting type of immune infiltration predominant in Cluster B (Figure 5A). We illustrate for the METABRIC and TCGA in Figure 5B and Supplementary Figure 8, how six immune cell types (Resting mast cells, Macrophages M2 and M1, T follicular helper, rested and activated memory T cells) have different distribution across all the clusters.

At last through generalized linear modelling, we illustrate how each immune cell type explains Cluster B vs Cluster A & C in Figure 5C.

We think these new results are important additions to the study which allows to better characterize the immune contexture of the here described clusters (see new paragraph of results p9).

8. Is Cluster B predictive of response to chemotherapy? There are publicly available datasets that can be leveraged to help address this.

This was another suggestion from the reviewer which lead to relevant new evidence. We therefore collected breast cancer gene expression data from neoadjuvant chemotherapy studies with pathological complete response (pCR) as an endpoint determined at surgery. We analyzed 8 cohorts for a total of 1377 samples.

We assigned a cluster to each sample using our single sample predictor and found that a very small proportion of the responders (pCR) were in Cluster B (11%). We therefore suggest in light of the poor prognosis associated with Cluster B that patients with pCR and in Cluster B may be at risk of relapse, despite the good initial response to neoadjuvant chemotherapy.

Interestingly 59% of the responders were found in Cluster C; the cluster with high immune infiltration and proliferation which is line with previous observations (PMID:28487444, PMID:20829329).

The remaining 30% of patients with pCR are in cluster A. These numbers support the observation from the survival analysis. We include Figure 4E showing the percentage of responders in each cluster, all this is described in a new paragraph of results end of p8 beginning of p9.

9. The single-sample classifier should be made publicly available as source code and ideally as a web service.

As suggested by the reviewer we provide all codes and data allowing us to reach our conclusions (hosted by github and attached in the source data file). We also provide a web service now available at: <http://eurostar.nebdal.no:5000/>. We believe this will encourage readers to try and score for immune clusters.

Minor comments:

1. There are numerous grammatical errors and typos that affect readability.

We hope the revised version is more readable and that grammatical errors are corrected

2. Figure 1 caption: the reference to “(A & C)” should be “(A & D)”.

We thank the reviewer for noticing this, it has been corrected.

Reviewer #2 (Remarks to the Author):

In this manuscript, Tekpli et. al. used patient tumor specimens and performed expression array of genes that are related to immune infiltration. They identified three clusters A, B, C, that show lymphocyte infiltration from low to high. They found that these three clusters exist in several sources of tumor samples including those from TCGA and METABRIC. They further showed that patients fall into Cluster B, which have intermediate lymphocyte infiltration, are associated with poor survival compared to Cluster A and C.

We thank the reviewer for the careful evaluation of our study and for the questions and comments which helped us improving our results.

The authors used a binomial logistic regression penalized by the lasso method and obtained a set of genes that was able to determine whether the patients belong to Cluster B. However, the false positive rate was $\sim 30\%$ ($=1 - 68.8\%$).

The accuracy of our single sample classifier when compared to the clustering method is illustrated by the ROC curve with AUC=85.8% (Figure 3A). Only 3.66% of the patients classified by clustering as "good prognosis" switched to the "bad prognosis" group using binomial regression. Our predictor has

therefore very low likelihood to assign patient with a relatively good prognosis in the bad prognosis cluster.

Then, ~30% of samples classified in the "bad prognosis" group by clustering appeared as "good prognosis" using the binomial analysis. This the reviewer refers to as "false positive rate". This observation leads us to conclude that the single sample predictor restricts the bad prognosis cluster to the cases with the worse prognosis. Indeed, we show in Supplementary Table 3 that the predictor generally improves the significant associations with survival as describe in page 6 of the new manuscript. We therefore believe that this stricter cluster assignment will reduce the number of false positive in future clinical applications.

Unsupervised clustering can be less reliable for small cohorts and samples in the 'grey' zone are difficult to assign to a cluster. The binomial logistic regression allowed us to learn from many samples (n=4546) what the clusters are 'made of' and to have a more homogeneous and reproducible grouping.

The predictor which allows to assign every next patient in the clinic to a cluster refines the belonging to Cluster B. We have now made the classifier available as a web service at: <http://eurostar.nebdal.no:5000/>

The authors also found that immune clusters are an independent prognostic factor in breast cancer, while the gene profiles of cluster B do not predict well on risk of recurrence (ROR).

Indeed, as noticed by the reviewer the immune clusters are an independent prognostic factor in breast cancer. We formally assessed the independency of the immune clusters from ROR scores using multivariable Cox regression analysis (Supplementary Table 5).

In addition, we now used net reclassification improvement (NRI) and integrated discrimination improvement (IDI) indices against the ROR scores to demonstrate the strength of the immune clusters as a new important biomarker in breast cancer. The results are presented in Supplementary Table 5 and described in the first half of p8.

We also tested the strength of the immune clusters against other known clinicopathological prognostic markers such as size, age, grade, stage and lymph node involvement (beginning of p7, Table 1 and Supplementary Table 4)

They further performed gene enrichment analysis and showed connections between immune Clusters and EMT signatures. However, the claim that "these results identify an EMT-stem-cell-like phenotype in Cluster B which associate with poor prognosis." does not hold from the data presented.

The main finding of our work is that we define immune clusters in breast cancer associated with specific immune cells. What kind of tumor cells and signatures these clusters will be associated with? We had no hypothesis for that before performing the analysis. The fact that the immune Cluster B captured strong EMT signals was interesting to us and worth reporting. We agree with the reviewers that not only the EMT signature was found in Cluster B. We therefore conducted several analyses to characterize the phenotypes and heterogeneity in Cluster B.

We first tested which gene set signatures were explaining Cluster B using generalized linear models, we found EMT, BOQUEST stem cell and LIM Mammary stem cell signatures as important features explaining

the phenotype underlying Cluster B (Figure 6C). However, these signatures were not sufficient to explain fully the poor prognosis associated with the immune-Cluster B (Supplementary Table 8), as alone these signatures were poorly associated with survival in univariate and multivariate cox regression analysis.

We also, show through new analysis that a pro-tumorigenic and resting immune contexture (Figure 5A, B and C) is an important factor explaining Cluster B.

Therefore, it is now clearer that EMT and stemness may only partly explain the poor prognosis in Cluster B.

Despite the dichotomy in Cluster B in regards to EMT vs proliferation (Figure 7 and end of p11, beginning of p 12), unifying factors are (i) medium type of immune infiltration which is pro-tumorigenic, (ii) the strong association with poor outcome across 15 breast cancer cohorts, (iii) association with response to neoadjuvant chemotherapy.

Essentially, Cluster A and Cluster C tumors from different collection sources showed more uniformed gene signatures, whereas Cluster B showed varied gene signatures among different tumor cohorts. In other words, enriched gene signatures in Cluster B are not consistent among different cohorts.

As pointed by the reviewer, a proliferative phenotype was also present in Cluster B in addition to EMT (Figure 6B and Supplementary Figure 9). The proliferative phenotype was however not retained as a defining feature of Cluster B when compared to Cluster A & C in generalized linear models, possibly because a proliferative phenotype is a dominating feature of Cluster C.

To assess the heterogeneity in Cluster B we performed unsupervised clustering using the gene set signatures. We divide Cluster B in two (i) with high EMT (Cluster B1) (ii) high proliferation (Cluster B2) (Figure 7A). In the two largest cohorts (METABRIC and TCGA), B2 cases with the proliferative phenotype had a worse outcome (Figure 7B and 7C). These analyses suggest that the heterogeneity in tumor phenotype within Cluster B may be relevant in regard to prognosis, it is however difficult for us to firmly conclude, as further dividing Cluster B only allow us to perform relevant statistical testing in the two largest cohorts.

The heterogeneity in Cluster B is now clearly defined and discussed in the manuscript (p11-12 of the results, p12-13 of the discussion).

Collectively, the reviewer finds that the study showed some interesting findings. However, the results are still at an observational stage and some of the conclusions are not justified.

We thank the reviewer for finding our study interesting, in light of the new results and analyses, we hope the reviewer will now find the results even more interesting, more mature and justified by clearer and well described data. We believe that through numerous validations, new clinical significance and further dissection of the immune cell types in the clusters our paper is beyond simply observational stage. In addition to what is stated above in the specific answers, we consider that among other things the following new results are worth noting:

- We further validate the immune clusters in 3 additional cohorts bringing the number of cohorts the cluster have been tested in to 15 and the number of samples to 6101.
- We clarify the clinical value and strength of the immune clusters as a new prognostic marker in breast cancer using several and complementary statistical tests.
- We provide a web service for our single sample predictor.
- We show the relevance of the immune clusters in regard to response to neoadjuvant chemotherapy.
- We clearly dissected the immune contexture associated with the clusters and found pro-tumorigenic and resting cell types (Macrophages M2, resting Mast cells and resting memory T cells) in the microenvironment of Cluster B samples.
- As we noted that the heterogeneity in gene signatures in Cluster B seemed to be a concern for the reviewer, we hope we now define it well in the revised version.

Reviewers' comments:

Reviewer #1 (Remarks to the Author):

Overall, the authors have done a nice job addressing my major comments from the prior round of review. As a result, their finding of Cluster B as a novel immune-associated subgroup with independent prognostic value in breast cancer is more strongly supported. The new analyses dissecting differences in immune composition between Clusters A, B, and C improve the manuscript, as do the new analyses linking Cluster B to inferior response to neoadjuvant therapy. The addition of software and an online tool for classifying breast cancer expression datasets into the three clusters will allow others to benefit from this work.

While the manuscript is vastly improved, there are still several key issues that should be addressed prior to publication.

Major:

1. The rate of pCR can differ as a function ER status and pam50 subtype. Have the authors examined whether Cluster B is associated with a lower response rate in breast tumors that are first stratified by ER status/pam50?
2. Page 9: why do the authors suggest that "responders in Cluster B may be candidates for testing of new neoadjuvant therapeutic options"? Perhaps the authors mean non-responders? Or perhaps they mean patients that are risk-classified to Cluster B prior to therapy?
3. Discussion: "We provide the first immune related prognostic factor in breast cancer". This is simply not true. Numerous studies have identified immunological correlates of survival in breast cancer. The authors should temper this claim along with any related claims in the paper.
4. The data in this manuscript do not support the title, "Immune contexture characterizes aggressive tumor phenotype through epithelial mesenchymal transition and stemness in breast cancer"? The key finding in this paper is Cluster B, a novel subtype that is characterized by (i) a distinct immune microenvironment and (ii) uniform representation of major clinical and molecular subtypes. The authors clearly show that EMT alone is weakly or insignificantly prognostic and is not restricted to Cluster B. Furthermore, stemness and cycling/proliferation gene sets often capture the same genes and the same biology, and these gene sets are also not restricted to Cluster B. Just because Cluster B captures tumors that are enriched for both "themes" does not imply that the clinical significance of Cluster B is uniquely driven by either of them. The authors should address this point throughout the manuscript.
5. The authors should ensure that the new RNA-seq expression data are made available as a TPM (or count) gene expression matrix with corresponding de-identified clinical data on the Gene Expression Omnibus. This is critical so that others can reproduce and extend the findings in this work without access control.

Minor:

1. It is unclear what the red text denotes in the survival analysis tables. In some instances, it appears to mark insignificant p-values (>0.05) and in others, significant p-values.
2. All supplementary tables should be provided as Excel spreadsheets.

Reviewer #2 (Remarks to the Author):

The revised manuscript described a unique immune profiling-based cluster, Cluster B, that shows prognostic values.

1. 3rd paragraph in Results "We then compared the transferability of the clustering obtained from FFPE (Nanostring)..." Please include results to show that "unsupervised clustering using the genes

found in all datasets in this study (n=509), we clustered the 104 MicMa-Agilent samples to obtain the same three clusters.”

2. Fig. 1A and Fig. 1D: the authors performed unsupervised clustering using different datasets and different platforms and observed three clusters. In addition to showing that Cluster B has intermediate lymphocyte infiltration shown in Fig. 1B, 1E, is there evidence to confirm that Cluster B in Fig. 1A and Fig. 1D represents the same type of cluster?

3. Please describe the calculation of lymphoid scores (y-axis unit of Fig. 1B, 1C) in Materials Section. What “set of genes’ markers of lymphocyte” was used and how the score was calculated.

4. Page 6, last paragraph, it reads: “It appeared that the lasso method decreased the number of samples in the poor prognosis group (Figure 3B).” Does it mean that “The Lasso method faithfully predicts samples to Cluster A & C, while a significant number of samples in Cluster B was assigned to Cluster A & C (Figure 3B)”.

5. Page 7: “Indeed, if we removed the immune clusters from the modelling, the Akaike Information Criterion (AIC) index was increased,” – Please include the results in supplemental data.

6. Fig. 7 B, 7C. What are the survival probabilities in Cluster A and Cluster C as compared to Cluster B1 and B2? Please plot the survival probabilities together and include it in Supplemental data. If Cluster B2 is most significantly associated with poor survival, Does Cluster B2 increase the prediction power in Fig. 3B?

7. Please submit Supplemental Tables as excel file format.

8. The manuscript requires extensive grammatical editing.

Reviewers' comments:

Reviewer #1:

Overall, the authors have done a nice job addressing my major comments from the prior round of review. As a result, their finding of Cluster B as a novel immune-associated subgroup with independent prognostic value in breast cancer is more strongly supported. The new analyses dissecting differences in immune composition between Clusters A, B, and C improve the manuscript, as do the new analyses linking Cluster B to inferior response to neoadjuvant therapy. The addition of software and an online tool for classifying breast cancer expression datasets into the three clusters will allow others to benefit from this work. While the manuscript is vastly improved, there are still several key issues that should be addressed prior to publication.

We thank the reviewer for her/his positive comments.

Major:

1. The rate of pCR can differ as a function ER status and pam50 subtype. Have the authors examined whether Cluster B is associated with a lower response rate in breast tumors that are first stratified by ER status/pam50?

We thank for the suggestion from the reviewer to look at response as a function of ER/PAM50 status. Accordingly, we analyzed the immune clusters in perspective of response to neoadjuvant chemotherapy in ER positive and ER negative cases separately. The results are now included in Supplementary Figure 8 and commented on p9 of the manuscript. Consistent with our observation that the immune Cluster B is a prognostic factor independent of the ER, we found that patients in Cluster B have a lower response rate in both in ER positive and ER negative cases. The same analysis is more difficult to perform with regard to PAM50 stratification, as Cluster B represents all PAM50 groups and the cohorts with response rate per cohort become very small when subdivided into both pCR, immune clusters and PAM50.

2. Page 9: why do the authors suggest that “responders in Cluster B may be candidates for testing of new neoadjuvant therapeutic options”? Perhaps the authors mean non-responders? Or perhaps they mean patients that are risk-classified to Cluster B prior to therapy?

We thank the reviewer for noticing this logistic lapse. We indeed meant "non-responders", i.e. as the current neo-adjuvant chemotherapies are not efficient for > 80% of the Cluster B patients' they should not be burdened with these and new strategies are needed for this group of patients. We modified the sentence (p9, end of 'Immune clusters and response to neoadjuvant chemotherapy').

3. Discussion: “We provide the first immune related prognostic factor in breast cancer”. This is simply not true. Numerous studies have identified immunological correlates of survival in breast cancer. The authors should temper this claim along with any related claims in the paper.

We followed the reviewer's advice and modified this sentence to read *"we provide a new immune-related subtype in breast cancer with relevance for prognosis and response to neoadjuvant chemotherapy in both ER positive and ER negative cases"*, see p12, beginning of discussion. We also reviewed the whole manuscript to temper similar claims.

4. The data in this manuscript do not support the title, "Immune contexture characterizes aggressive tumor phenotype through epithelial mesenchymal transition and stemness in breast cancer"? The key finding in this paper is Cluster B, a novel subtype that is characterized by (i) a distinct immune microenvironment and (ii) uniform representation of major clinical and molecular subtypes. The authors clearly show that EMT alone is weakly or insignificantly prognostic and is not restricted to Cluster B. Furthermore, stemness and cycling/proliferation gene sets often capture the same genes and the same biology, and these gene sets are also not restricted to Cluster B. Just because Cluster B captures tumors that are enriched for both "themes" does not imply that the clinical significance of Cluster B is uniquely driven by either of them. The authors should address this point throughout the manuscript.

We fully agree with the reviewer, we should have changed the title of the manuscript earlier, after the first revision. We now paraphrase the excellent summary of the reviewer and opted for the title "An independent prognostic and predictive subtype of breast cancer with distinct immune microenvironment", which we think fits our study better.

We also now clearly describe how Cluster A, B and C are associated to proliferation and EMT and emphasize that while these phenotypes are associated with the clusters, they may only be relevant to subdivide cluster B into B1 and B2. One of the sentences summarizing well our conclusion regarding these phenotypes is: "Our results suggest that only when accompanied by a certain immune contexture the EMT or the proliferative phenotype result in poor prognosis" (p11).

5. The authors should ensure that the new RNA-seq expression data are made available as a TPM (or count) gene expression matrix with corresponding de-identified clinical data on the Gene Expression Omnibus. This is critical so that others can reproduce and extend the findings in this work without access control.

We uploaded the FPKM RNA-seq gene expression data at GEO as suggested GSE135298.

To review GEO accession GSE135298:

Go to <https://www.ncbi.nlm.nih.gov/geo/query/acc.cgi?acc=GSE135298>

Enter token `wnazkqeanpkbfuv` into the box

The information is now available in the revised manuscript.

Minor:

1. It is unclear what the red text denotes in the survival analysis tables. In some instances, it appears to mark insignificant p-values (>0.05) and in others, significant p-values.

The previous colors were just a way for us to navigate the data quickly, and we agree this was confusing for the readers. Only p-values < 0.05 are now in bold.

2. All supplementary tables should be provided as Excel spreadsheets.

All Supplementary Tables have now been uploaded as .xlsx.

Reviewer #2:

The revised manuscript described a unique immune profiling-based cluster, Cluster B, that shows prognostic values.

1. 3rd paragraph in Results “We then compared the transferability of the clustering obtained from FFPE (Nanostring)...”. Please include results to show that “unsupervised clustering using the genes found in all datasets in this study (n=509), we clustered the 104 MicMa-Agilent samples to obtain the same three clusters.”

As suggested by the reviewer we included the two-way table and the Fisher exact test p-value showing that the results of the clustering using either the 760 genes or the 509 genes are significantly concordant (Supplementary Table 1).

2. Fig. 1A and Fig. 1D: the authors performed unsupervised clustering using different datasets and different platforms and observed three clusters. In addition to showing that Cluster B has intermediate lymphocyte infiltration shown in Fig. 1B, 1E, is there evidence to confirm that Cluster B in Fig. 1A and Fig. 1D represents the same type of cluster?

We believe indeed that through the manuscript we find several characteristics showing that the immune clusters obtained from different datasets share common features in addition to the gradual lymphocyte infiltration.

1. We find an increased infiltration of Macrophages M2 and Mast cells in Cluster B (Figure 5A).
2. Are strongly associated with survival (Table 1)
3. Associate with response to chemotherapy (Figure 4E)
4. The GSVA pathway analysis shows a similar distribution of the EMT/proliferation phenotype across the clusters (Figure 6B)

3. Please describe the calculation of lymphoid scores (y-axis unit of Fig. 1B, 1C) in Materials Section. What “set of genes’ markers of lymphocyte” was used and how the score was calculated.

We apologize that this information was not easy to find in our previous version of the Manuscript, we have a section in the Material and Methods p17 entitled ‘*Nanodissect analysis, lymphoid and myeloid scores*’ in which you will find all the necessary information to calculate the lymphoid scores given the gene set in Supplementary Table 10.

4. Page 6, last paragraph, it reads: “It appeared that the lasso method decreased the number of samples in the poor prognosis group (Figure 3B).” Does it mean that “The Lasso method faithfully predicts samples to Cluster A & C, while a significant number of samples in Cluster B was assigned to Cluster A & C (Figure 3B)”.

Yes, the reviewer understood correctly, 96% of the samples assigned to Cluster A & C by the clustering method were also assigned to Cluster A & C by the lasso method while 69% of the samples assigned to Cluster B using the clustering method were assigned to Cluster B using lasso. We could have found the same rate of 'errors' in the two classes and we thought this skewness was worth noticing, however, our lasso method remains highly sensitive and specific with an AUC == 85.8%. In addition, as stated in the manuscript (end of p6 - beginning of p7), the lasso method refined the immune clusters as the log rank p values for survival / Kaplan Meier analysis were lower when calculated upon the lasso assignments (Supplementary Table 4). Therefore, we believe that learning the cluster assignments from the initial clustering of several cohorts helps us using lasso to obtain a homogene and refined grouping.

5. Page 7: "Indeed, if we removed the immune clusters from the modelling, the Akaike Information Criterion (AIC) index was increased," – Please include the results in supplemental data.

The AIC indexes from the multivariable cox regression analyses for each cohort with all variables and without the immune clusters are now available in Supplementary Table 6

6. Fig. 7 B, 7C. What are the survival probabilities in Cluster A and Cluster C as compared to Cluster B1 and B2? Please plot the survival probabilities together and include it in Supplemental data. If Cluster B2 is most significantly associated with poor survival, Does Cluster B2 increase the prediction power in Fig. 3B?

As suggested, by the reviewer for the two largest cohorts METABRIC and TCGA, we plotted survival probabilities for Cluster A, B1, B2 and C. Indeed, it seems that for these two cohorts the patients in B2 have a worse survival. While this may be interesting for further analysis and follow up, we indicate to readers and reviewers that we are underpowered to firmly conclude on the clinical relevance of these sub-clusters (manuscript p12, last paragraphs of results). Indeed, splitting Cluster B to B1 and B2 results in small groups for many cohorts and further affirming that B2 patients will have the worse survival in all cohorts is not possible.

We appreciate the suggestion of the reviewer to use this information for further refining our lasso prediction but before starting on this rather computationally intensive task we need to figure out the clinical relevance of these subclusters which is not entirely clear yet.

7. Please submit Supplemental Tables as excel file format.

All Supplementary Tables have now been uploaded as .xlsx.

8. The manuscript requires extensive grammatical editing.

While revising the manuscript we corrected all grammatical errors

Reviewers' comments:

Reviewer #1 (Remarks to the Author):

Tekpli and colleagues have satisfactorily addressed major comments from the last round of review. However, several minor concerns remain.

1. While the authors have shown that responders to neoadjuvant therapy are depleted in Cluster B, the potential clinical utility of Cluster B as a predictive biomarker of non-response remains unclear. To more thoroughly address this question, the authors should split by pCR and non-pCR and show a significant difference in patients assigned to Cluster B, both for all patients and for ER+ and ER- patients separately within each cohort. The authors should also consider evaluating the predictive performance of Cluster B via ROC analysis, where response vs. non-response is measured as a continuous function of the probability of assignment to Cluster B (from the logistic regression model), and an optimal split defined and tested across held-out cohorts. In all cases, sensitivity, specificity, positive predictive value, and negative predictive value should be reported. If the authors elect to skip such analyses, strong claims about the predictive value of Cluster B should be avoided/tempered.
2. Related to the above, I would suggest removing "predictive" from the title. A more appropriate title might be: "A poor-prognosis subtype of breast cancer defined by a distinct tumor immune microenvironment" or "An independent subtype of breast cancer defined by an aggressive tumor phenotype and a distinct immune microenvironment".
3. To encourage broad adoption of Cluster B, the authors might consider assigning it a descriptive name.
4. The concluding sentence in the abstract is not justified by the data presented in this manuscript. What "reciprocity" are the authors referring to? Do the authors instead mean "association"?
5. "More global approaches are needed to establish the role of the immune contexture in breast cancer." Despite the identification of Cluster B, this manuscript does not introduce a novel "global approach" for studying immune contexture.
6. "The herein described immune clusters are dependent on quality rather than quantity of the immune infiltration and are independent of other prognostic factors, including PAM50." This is not supported by the presented data. Rather, the "...immune clusters are dependent on both the abundance and composition of the immune infiltrate...". Implying otherwise is inconsistent with figures showing that Clusters A, B, and C reflect differences in total immune content and differences in leukocyte composition.

Reviewer #2 (Remarks to the Author):

Specific comments:

1. Page 6: "We found that 96.3% of the samples assigned to Cluster A & C by clustering were predicted to be A & C by the model while 68.8% of the samples assigned to Cluster B through clustering were found in Cluster B using the lasso method." Does the data refer to Fig. 3B? If so, please cite Fig. 3B at the end of the statement.
2. The reviewer recommends the authors to add a panel in Fig. 4 to show the findings described in the last three paragraphs in the section "Immune clusters are an independent prognostic factor in breast cancer".
3. Page 9, 1st paragraph in "Immune clusters and response to neoadjuvant chemotherapy": The authors need to indicate the source of the studies and include the gene expression and pCR data used to generate Fig. 4E.

4. Fig. 5B and 5C are mislabeled. Fig. 5C should be Fig. 5B. In current Fig. 5B, how did the authors combine Clusters A & C? Did the authors take the average enrichment of immune cell types of Cluster A & C and compare the values to that of Cluster B? If so, it may skew the data to one dominant cluster. Regardless, the authors should plot HR of immune cell types comparing Cluster B to Cluster A and Cluster B to Cluster C individually. Similarly, as regard to Fig. 6C, the authors should perform the comparison of HR on gene signatures between Cluster B to Cluster A and Cluster B to Cluster C.

5. Page 10: "To further characterize the phenotype associated with the poor prognosis in Cluster B, we identified through differential gene expression analysis the genes significantly overexpressed in Cluster B. We found 909 genes upregulated in Cluster B when compared to Cluster A and Cluster C separately (Bonferroni corrected p-value < 0.0001)". Please include in a supplementary excel file the values of differentially expressed genes in Cluster B, Cluster A, Cluster C, and A/C combined.

6. Page 10: "To further characterize the relationship between the immune clusters and cancer cell phenotype, we curated gene sets for various pathways related to EMT, stem-cells, hypoxia and proliferation (Supplementary Table 9)." Please cite references where the curated gene sets were obtained.

7. Page 10 last sentence: "Unsupervised clustering of averaged-gene-set scores almost perfectly rediscovered the immune clusters (Figure 6B) demonstrating an association between immuneclusters and the stem cells / EMT-related gene signatures." This is an overstatement and is not correct. As seen on the plot, there are two unsupervised clusters. Samples in cluster B are separately merged to Cluster A as well as Cluster C.

8. Page 8: The 2nd from last paragraph "Multivariable regression analysis ..." is difficult to follow and need to be better explained. For the statement "Multivariable regression analysis confirmed that the immune clusters bring additional prognostic value to the ROR scores (Supplementary Table 7)", does this refer to the top panel of Supplementary Table 7? If so, it looked as though ROR shows much better HR scores. For the NRI and IDI analyses, were the immune cluster alone compared to ROR or the combined immune cluster and ROR compared to ROR alone? If the former, it would be interesting to show whether combining the immune cluster and ROR increases the prognostic value. If the latter, the data in Supplementary Table 7 implies that adding immune cluster to ROR does not improve much on prognostic value.

9. Page 12: Discussion "The herein described immune clusters are dependent on quality rather than quantity of the immune infiltration". This is not accurate. None of the data in this manuscript compared quality and quantity of the immune infiltration.

10. Page 14: "Our study demonstrates using more recent algorithms that the EMT phenotype is not confined to basal-like tumors, ..., giving further support to the novelty of our results." The statements in this paragraph are not properly made. It was known that basal-like tumors have enhance EMT features, but similar to Cluster B, not all basal-like tumors have the EMT phenotype. No evidence in this manuscript shows better EMT association compared to basal-like or claudin-low tumors. Furthermore, while the authors did not include the claudin-low data, the lack of correlation between Claudin-low and Cluster B does not prove the novelty of current study.

11. The manuscript provides very interesting results. The reviewer recommends the authors to improve the organization of the text and the Figures. Many short and one sentence paragraphs exist throughout the manuscript. Figures could also be combined to make them cohesive - eg. Fig. 7 is not quite sufficient to stand alone as a figure.

Minor:

1. The authors should edit the phrase: "It appeared that the lasso method decreased the number of samples in the poor prognosis group (Figure 3B)." It is unclear what the poor prognosis group was referred to.
2. Page 8. The authors need to indicate that ROR was generated using PAM50. They should also refer back to Fig. 1 and Fig. 4B that immune clusters do not overlap with PAM50 scores.
3. Fig. 5B: Is the HR presented as a log2 value in x-axis? Why negative values?
4. Fig. 5C, label x axis, indicate color representations.
5. Fig. 6C: The HR value should > 0 ? Why plot the x-axis from -1?
6. Table 1 needs to be shown in an excel format.
7. Typo in Supplementary Fig. S7C: It should be neg not meg

Reviewers' comments:

Reviewer #1 (Remarks to the Author):

Tekpli and colleagues have satisfactorily addressed major comments from the last round of review. However, several minor concerns remain.

1. While the authors have shown that responders to neoadjuvant therapy are depleted in Cluster B, the potential clinical utility of Cluster B as a predictive biomarker of non-response remains unclear. To more thoroughly address this question, the authors should split by pCR and non-pCR and show a significant difference in patients assigned to Cluster B, both for all patients and for ER+ and ER- patients separately within each cohort. The authors should also consider evaluating the predictive performance of Cluster B via ROC analysis, where response vs. non-response is measured as a continuous function of the probability of assignment to Cluster B (from the logistic regression model), and an optimal split defined and tested across held-out cohorts. In all cases, sensitivity, specificity, positive predictive value, and negative predictive value should be reported. If the authors elect to skip such analyses, strong claims about the predictive value of Cluster B should be avoided/tempered.

We agree with the reviewer that more analyses are required to claim that the immune clusters are predictive of response. We therefore assessed in each cohort the distribution (chi-square p-values, Supplementary Table 8) of the pCR and non-pCR cases across the immune clusters taking in account whole cohorts, ER positive and ER negative cases independently. When considering the whole cohort, we found the distribution of the responders significantly different across immune clusters, with less responders in Cluster B and most responders in Cluster C. When splitting by ER status the same tendency was observed although not always significant.

We, in fact, brought the topic of treatment response (prediction) into the paper based on the reviewer's suggestion and it was very positive that we saw associations. However, we agree that our evidence is not strong enough to highlight prediction to treatment response in the title. This requires a controlled experiment, treatment type dependent and with stricter measurement of response than the one we have today. Therefore, following the reviewer's suggestion to assess the predictive value of the clusters with ROC curve based on the treatment response data available to us now did not bring strong signal.

Our interpretation is that the association of the immune clusters with response is significant not only because of the negative contribution of cluster B but also because of the positive contribution of cluster C. When tested separately against the rest these two clusters alone did not project in strong enough signals with the given number of patients.

In light of these new set of observations, we decided to temper the 'predictive' wording and used 'association' instead.

2. Related to the above, I would suggest removing "predictive" from the title. A more appropriate title might be: "A poor-prognosis subtype of breast cancer defined by a distinct tumor immune microenvironment" or "An independent subtype of breast cancer defined by an aggressive tumor phenotype and a distinct immune microenvironment".

In light of the analysis performed as stated above, we opted for the title: "An independent poor-prognosis subtype of breast cancer defined by a distinct tumor immune microenvironment"

3. To encourage broad adoption of Cluster B, the authors might consider assigning it a descriptive name. We have been thinking at the suggestion of the reviewer and tried to introduce at several places in the manuscript that Cluster B is THE pro-tumorigenic immune infiltration Cluster or THE poor prognosis cluster.

4. The concluding sentence in the abstract is not justified by the data presented in this manuscript. What “reciprocity” are the authors referring to? Do the authors instead mean “association”?

We indeed meant association / correlation and changed the word reciprocity for correlation as it supported by the data of Fig.6D in which EMT correlates with M2 Macrophages while proliferation correlates with M1-Macrophages.

5. "More global approaches are needed to establish the role of the immune contexture in breast cancer." Despite the identification of Cluster B, this manuscript does not introduce a novel “global approach” for studying immune contexture.

We changed the sentence pointed at by the reviewer for: “More studies are needed to specify the role and the clinical relevance of the immune contexture in breast cancer”

6. "The herein described immune clusters are dependent on quality rather than quantity of the immune infiltration and are independent of other prognostic factors, including PAM50." This is not supported by the presented data. Rather, the “...immune clusters are dependent on both the abundance and composition of the immune infiltrate...”. Implying otherwise is inconsistent with figures showing that Clusters A, B, and C reflect differences in total immune content and differences in leukocyte composition.

We adopted the formulation suggested by the reviewer: “The herein described immune clusters are dependent on both the abundance and composition of the immune infiltrate and are independent of other prognostic factors, including PAM50”

Reviewer #2 (Remarks to the Author):

Specific comments:

1. Page 6: “We found that 96.3% of the samples assigned to Cluster A & C by clustering were predicted to be A & C by the model while 68.8% of the samples assigned to Cluster B through clustering were found in Cluster B using the lasso method.” Does the data refer to Fig. 3B? If so, please cite Fig. 3B at the end of the statement.

Yes indeed, this statement is related to Fig3B, so we added the reference of the Figure as suggested.

2. The reviewer recommends the authors to add a panel in Fig. 4 to show the findings described in the last three paragraphs in the section “Immune clusters are an independent prognostic factor in breast cancer”.

We think it is difficult to use Figures to illustrate the results of the Multivariable cox regression analysis. While we have supplementary Figure 6 to illustrate our purpose in these paragraphs, we have Table 1

which we believe is the best and full description of the data.

3. Page 9, 1st paragraph in “Immune clusters and response to neoadjuvant chemotherapy”: The authors need to indicate the source of the studies and include the gene expression and pCR data used to generate Fig. 4E.

The gene expression data for the cohort with pCR are pointed at in Supplementary Table 2. All datasets are downloadable at the GEO IDs indicated with the associated pCR.

4. Fig. 5B and 5C are mislabeled. Fig. 5C should be Fig. 5B. In current Fig. 5B, how did the authors combine Clusters A & C? Did the authors take the average enrichment of immune cell types of Cluster A & C and compare the values to that of Cluster B? If so, it may skew the data to one dominant cluster. Regardless, the authors should plot HR of immune cell types comparing Cluster B to Cluster A and Cluster B to Cluster C individually. Similarly, as regard to Fig. 6C, the authors should perform the comparison of HR on gene signatures between Cluster B to Cluster A and Cluster B to Cluster C. We thank the reviewer for this comment. Indeed, in the generalized model analysis we compared cluster B to the rest (Cluster A & C as one group) as we were seeking to identify the immune cell types (or pathways) characterizing Cluster B. We think this analysis is sound to highlights the feature specific of the poor prognosis subtype (Cluster B) versus the 'rest' (Cluster A & C). But we agree with the reviewer that comparing Cluster B to A and C separately is valuable. We added the analysis suggested by the reviewer in Supplementary Figure 10 and Supplementary Figure 12 and found that the pathways and immune cell types characterizing Cluster B when compared to Cluster A & C were mostly recapitulated when looking at (i) Cluster B versus Cluster A and (ii) Cluster B versus Cluster C separately. Important to notice that this comparison of immune cell types is more difficult in the case of comparing Cluster B to A as cluster A is extremely poor in immune cells to begin with.

5. Page 10: “To further characterize the phenotype associated with the poor prognosis in Cluster B, we identified through differential gene expression analysis the genes significantly overexpressed in Cluster B. We found 909 genes upregulated in Cluster B when compared to Cluster A and Cluster C separately (Bonferroni corrected p-value < 0.0001).” Please include in a supplementary excel file the values of differentially expressed genes in Cluster B, Cluster A, Cluster C, and A/C combined.

The results are added in Supplementary Table 10

6. Page 10: “To further characterize the relationship between the immune clusters and cancer cell phenotype, we curated gene sets for various pathways related to EMT, stem-cells, hypoxia and proliferation (Supplementary Table 9).” Please cite references where the curated gene sets were obtained.

We now clearly state in the text that the curated sets are from the MsigDB with an additional set coming from PMID: 25214461.

7. Page 10 last sentence: “Unsupervised clustering of averaged-gene-set scores almost perfectly rediscovered the immune clusters (Figure 6B) demonstrating an association between immune clusters and the stem cells / EMT-related gene signatures.” This is an overstatement and is not correct. As seen on the plot, there are two unsupervised clusters. Samples in cluster B are separately merged to Cluster A as well as Cluster C.

We thank the reviewer for this observation and modified our sentence accordingly: “Unsupervised clustering of averaged-gene-set scores clearly separated the immune clusters A and C, while Cluster B was divided in two (Figure 6B). These results suggested an association between immune-clusters and the stem cells / EMT-related gene signatures.”

8. Page 8: The 2nd from last paragraph “Multivariable regression analysis ...” is difficult to follow and need to be better explained. For the statement “Multivariable regression analysis confirmed that the immune clusters bring additional prognostic value to the ROR scores (Supplementary Table 7)”, does this refer to the top panel of Supplementary Table 7? If so, it looked as though ROR shows much better HR scores. For the NRI and IDI analyses, were the immune cluster alone compared to ROR or the combined immune cluster and ROR compared to ROR alone? If the former, it would be interesting to show whether combining the immune cluster and ROR increases the prognostic value. If the latter, the data in Supplementary Table 7 implies that adding immune cluster to ROR does not improve much on prognostic value.

We tried to improve the description of this paragraph and we understand that suppl table 7 has created confusion. The main result showing the independence of the immune cluster as a prognostic factor is shown in table 1, where the immune cluster are included in the multivariate analysis together with the PAM50 subtypes and other known clinical prognostic parameters. Because the ROR score (as in Prosigna or as computed in Parker et al., JCO 2009) is a boosted and summarized version of the PAM50 signal, we were advised in the review process of this paper to also assess the independence of the immune cluster towards ROR. We did that and again saw that the immune cluster still remains independent prognostic factor as denoted by the significant p values in the multivariable analysis of Suppl Table 7).

Of course we cannot beat the ROR (do not expect to get stronger p-values or HR than those for ROR) as the latter is the synthesis of all we know in breast cancer about prognosis, but the immune clusters B still remains significant in most cohorts (9 out of 11) a source of additional information on prognosis.

Concerning the NRI IDI analysis, this was also added on the suggestion of reviewer 1 and was found satisfactory at the previous round of review. This analysis assesses how immune clusters combined with ROR scores would help to better classify patients according to survival when compared to ROR scores alone. The fact that for each cohort, we found positive (and significant) NRI and IDI in the combined case shows again (with other analysis included in the study) that the immune clusters are an independent prognostic factor and add to the prognostic value of PAM50 / ROR scores. We agree with the reviewer that the effect (NRI and IDI values) may not be strong in this setting, but we still believe that it may be of importance for an individual patient and when considering particular treatments.

9. Page 12: Discussion “The herein described immune clusters are dependent on quality rather than quantity of the immune infiltration”. This is not accurate. None of the data in this manuscript compared quality and quantity of the immune infiltration.

We thank the reviewer for this comment, we modified this sentence for: “The herein described immune clusters are dependent on both the abundance and composition of the immune infiltrate and are independent of other prognostic factors, including PAM50”

10. Page 14: “Our study demonstrates using more recent algorithms that the EMT phenotype is not confined to basal-like tumors, ..., giving further support to the novelty of our results.” The statements in

this paragraph are not properly made. It was known that basal-like tumors have enhance EMT features, but similar to Cluster B, not all basal-like tumors have the EMT phenotype. No evidence in this manuscript shows better EMT association compared to basal-like or claudin-low tumors. Furthermore, while the authors did not include the claudin-low data, the lack of correlation between Claudin-low and Cluster B does not prove the novelty of current study.

We reviewed this part of the discussion to now read:” Our study shows using recent algorithms that the EMT phenotype is enriched in Cluster B. In breast cancer, a recent gene transcriptional profiling has identified an EMT gene-expression signature associated with claudin-low and metaplastic breast cancers⁴⁷. However, the claudin-low subtype in the METABRIC cohort did not correlate with Cluster B.”

11. The manuscript provides very interesting results. The reviewer recommends the authors to improve the organization of the text and the Figures. Many short and one sentence paragraphs exist throughout the manuscript. Figures could also be combined to make them cohesive - eg. Fig. 7 is not quite sufficient to stand alone as a figure.

We thank the reviewer for his positive comment. Following it, we reorganize the Figure and sentences to have less short paragraph and Figure 7 has been merged with Figure 6. We hope to have further improvements if necessary, with copyeditors of Nature Communications may this be necessary.

Minor:

1. The authors should edit the phrase: “It appeared that the lasso method decreased the number of samples in the poor prognosis group (Figure 3B).” It is unclear what the poor prognosis group was referred to.

done

2. Page 8. The authors need to indicate that ROR was generated using PAM50. They should also refer back to Fig. 1 and Fig. 4B that immune clusters do not overlap with PAM50 scores.

done

3. Fig. 5B: Is the HR presented as a log2 value in x-axis? Why negative values?

Using the binomial glm (generalized linear models) method can result in negative coefficient (HR / estimates) for the variable in the fitted model. It means that the variable as a strong negative impact on explaining in our case Cluster A&C. We however, replace the HR to Estimates in the Figures to avoid confusions and is more appropriate to binomial generalized models.

4. Fig. 5C, label x axis, indicate color representations.

done

5. Fig. 6C: The HR value should > 0? Why plot the x-axis from -1?

Changed to start at 0

6. Table 1 needs to be shown in an excel format.

Provided

7. Typo in Supplementary Fig. S7C: It should be neg not meg
Corrected

REVIEWERS' COMMENTS:

Reviewer #1 (Remarks to the Author):

The authors have satisfactorily addressed my concerns from the last round of review. Before publication, I suggest the authors tone down their statements linking their findings to stem cell phenotypes throughout the manuscript, particularly in the title of Figure 6, which suggests "two mutually exclusive stem cell states in breast cancer". Such associations have not been adequately substantiated in this work.

Reviewer #2 (Remarks to the Author):

Fig. 6C far right, one of the estimates (95% CI), possibly HALLMARK EMT, is missing.

REVIEWERS' COMMENTS:

Reviewer #1 (Remarks to the Author):

The authors have satisfactorily addressed my concerns from the last round of review. Before publication, I suggest the authors tone down their statements linking their findings to stem cell phenotypes throughout the manuscript, particularly in the title of Figure 6, which suggests “two mutually exclusive stem cell states in breast cancer”. Such associations have not been adequately substantiated in this work.

We have corrected the title of Figure 6.

Reviewer #2 (Remarks to the Author):

Fig. 6C far right, one of the estimates (95% CI), possibly HALLMARK EMT, is missing.

We thank the reviewer for noticing this, it has been corrected.